# Improving black-box optimization in VAE latent space using decoder uncertainty

**Pascal Notin**
Department of Computer Science
University of Oxford
Oxford, UK
`pascal.notin@cs.ox.ac.uk`

**José Miguel Hernández-Lobato**
Department of Engineering
University of Cambridge
Cambridge, UK
`jmh233@cam.ac.uk`

**Yarin Gal**
Department of Computer Science
University of Oxford
Oxford, UK
`yarin@cs.ox.ac.uk`

## Abstract

Optimization in the latent space of variational autoencoders is a promising approach to generate high-dimensional discrete objects that maximize an expensive black-box property (e.g., drug-likeness in molecular generation, function approximation with arithmetic expressions). However, existing methods lack robustness as they may decide to explore areas of the latent space for which no data was available during training and where the decoder can be unreliable, leading to the generation of unrealistic or invalid objects. We propose to leverage the epistemic uncertainty of the decoder to guide the optimization process. This is not trivial though, as a naive estimation of uncertainty in the high-dimensional and structured settings we consider would result in high estimator variance. To solve this problem, we introduce an importance sampling-based estimator that provides more robust estimates of epistemic uncertainty. Our uncertainty-guided optimization approach does not require modifications of the model architecture nor the training process. It produces samples with a better trade-off between black-box objective and validity of the generated samples, sometimes improving both simultaneously. We illustrate these advantages across several experimental settings in digit generation, arithmetic expression approximation and molecule generation for drug design.

## 1 Introduction

We consider the task of optimizing an expensive black-box objective function taking inputs in a *high-dimensional discrete* space. This could be for example finding new molecules for drug design, or automatically generating a computer program that matches a desired output. Solving this task directly in the original space (e.g., with discrete local search methods such as genetic algorithms) may be challenging given the complex structure and high dimensionality of the data. Recently, Variational autoencoders (VAEs) [1, 2] have been successfully leveraged to model a wide range of discrete data modalities — from natural language [3], to arithmetic expressions [4], computer programs [5] or molecules [6]. By learning a lower-dimensional continuous representation of objects in their latent space, VAEs allow to transform the original discrete optimization problem into a simpler *continuous* optimization one in latent space. For example, this can be achieved via Bayesian Optimization in the latent space, or via gradient ascent with a jointly-trained neural network predicting the black box

35th Conference on Neural Information Processing Systems (NeurIPS 2021).

property from the latent space representation [6, 7]. Initial methods in this area have suffered from the fact that the search in latent space may explore areas for which no data was available at train time, and therefore where the decoder network of the VAE will be unreliable [8]: seemingly good candidate points in latent space may be decoded into objects that are invalid, unrealistic or low quality.

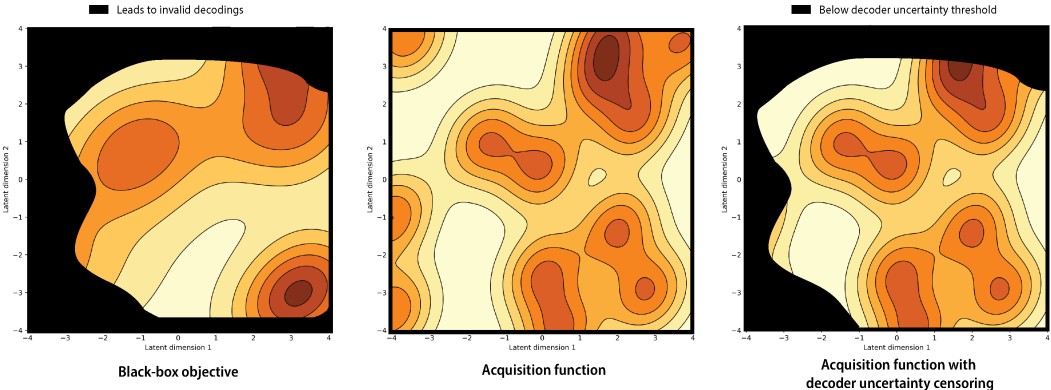

Figure 1: **Uncertainty-guided optimization in VAE latent space** The goal of black-box optimization in latent space is to attain regions with high values of the back-box objective after decoding, while avoiding the regions that lead to invalid decodings (left). Standard Bayesian Optimization in latent space may query these suboptimal areas (e.g., regions on left hand side, center). High decoder uncertainty regions overlap with regions leading to invalid decodings (right), so that censoring high uncertainty points helps guiding the optimization towards the most promising latent points.

While several methods have been introduced to promote validity of decoded objects (§2.1), they either focus on modifying the generative model learning procedure or adapting the decoder architecture to satisfy the syntactic requirements of the data modality of interest. We propose instead to quantify and leverage the uncertainty of the decoder network to guide the optimization process in latent space (Fig. 1). This approach does not require any change to the model training nor architecture, and can easily be integrated within several optimization frameworks. It results in a better trade-off between the values of the black-box objective and the validity of the newly generated objects, sometimes improving both simultaneously.

To be effective, this method requires robust estimates of model uncertainty for high dimensional structured data. Existing methods for uncertainty estimation in this domain often rely on heuristics or make independence assumptions to make computations tractable (§2.2). We demonstrate that such assumptions are not appropriate in our setting, and propose new methods for uncertainty estimation in high dimensional structured data instead.

Our contributions are as follows:

- We introduce an algorithm to quantify the uncertainty of high-dimensional discrete data, and use it to estimate the uncertainty of the decoder (§3);
- We show how the uncertainty of the decoder can be incorporated across several optimization frameworks, including gradient ascent and Bayesian optimization (§4);
- We illustrate our approach in a digit generation setting — a simple setup to provide intuition for the method — then quantify its benefits in the more complex tasks of arithmetic expressions approximation and molecule generation, covering a diverse set of decoder architectures across experiments (Convolutional, Recurrent and Graph Neural Networks) (§5).

## 2 Background

### 2.1 Optimization of high-dimensional discrete objects with generative models

Focusing on molecular generation, Gómez-Bombarelli et al. [6] propose to train a VAE to learn a distribution over the so-called SMILES representation of molecules (ie. linear sequences of characters) [9], and subsequently perform the optimization in the latent space. Since the SMILES

representation follows strict syntactic requirements that are not explicitly enforced by the generative model, promising points in latent space may be decoded into invalid molecules. To improve the validity of decoded sequences, Kusner et al. [4] and Dai et al. [5] develop task-specific grammar rules into the VAE decoder, focusing on use cases in molecular and computer program generation. However, crafting the corresponding rules requires domain-specific knowledge, needs to be designed from scratch for each new task, and may not be straightforward to elicit in the first place (e.g., digit generation example in 5.1). Griffiths and Hernández-Lobato [10] and Liu et al. [11] propose instead to formulate the problem as a constrained Bayesian Optimization and chance-constrained optimization task respectively to simultaneously optimize the target property as well as the probability to generate valid sequences. These two approaches require nonetheless access to a function that quantifies the validity or realism of objects in the training data, which is not readily available in many practical applications. A different line of research has focused on representing high-dimensional structured objects as graphs instead [12, 13]. The Junction Tree VAE (JT-VAE) [14] generates systematically valid molecular graphs, by first generating a tree-structured scaffold over a finite set of molecular clusters, and then assembling these clusters back into molecules with a message passing network. The MolDQN [15] casts the optimization problem as a reinforcement learning task (double Q-learning), which allows in turn to more naturally extend to simultaneous optimization of different objectives. GraphAF [16] combines the strengths of autoregressive and flow-based approaches to efficiently generate realistic and valid molecular graphs. Lastly, Tripp et al. [17] show that the black-box optimization performance can be further enhanced by iteratively retraining the generative model on the points selected during optimization, with weights that depend on their objective function value.

Our approach deviates from all the above in that it is is representation-agnostic (works with sequences or graphs), does not require domain-knowledge to craft custom rules or constraints, does not change the model architecture nor the learning procedure and can be combined with several of these approaches to reach even stronger optimization performance (see § 5).

## 2.2 Quantifying model uncertainty

Adopting a Bayesian viewpoint, the overall uncertainty of a model in a given region of the input space can be broken down into two types of uncertainty [18]:

- **Epistemic uncertainty:** Uncertainty due to lack of knowledge about that particular region of the input space — the posterior predictive distribution is broad in that region due to lack of information that can be reduced by collecting more data;

- **Aleatoric uncertainty:** Uncertainty due to inherent stochasticity/noise in the observations in that region — collecting additional data would not further reduce that uncertainty.

We denote input points as $x$, outputs as $y$ and the training data as $\mathcal{D}$. The total uncertainty $\mathcal{U}$ of a model at an input point $x$ is typically measured by the predictive entropy, ie. the entropy of the predictive posterior distribution $P(y|x, \mathcal{D})$:

$$\mathcal{U}(x) = \mathcal{H}(P(y|x, \mathcal{D})) = \sum_y -P(y|x, \mathcal{D}) \log P(y|x, \mathcal{D}) dy. \tag{1}$$

Denoting $P(\theta|\mathcal{D})$ the posterior distribution over model parameters $\theta$, we can further decompose the predictive entropy $\mathcal{U}$ as the sum of two terms:

$$\mathcal{U}(x) = \underbrace{(\mathcal{H}(P(y|x, \mathcal{D})) - \mathbb{E}_{P(\theta|\mathcal{D})}(\mathcal{H}(P(y|x, \theta))))}_{\text{Mutual Information } \mathcal{M}} + \underbrace{\mathbb{E}_{P(\theta|\mathcal{D})}(\mathcal{H}(P(y|x, \theta)))}_{\text{Expected entropy } \mathcal{E}} \tag{2}$$

The first term — the Mutual Information $\mathcal{M}$ between model parameters $\theta$ and the prediction $y$ — is a measure of epistemic uncertainty, as it quantifies the magnitude of the change in model parameters that would result from observing $y$. If the model is uncertain about its prediction for $y$, the change in model coefficients from observing $y$ should be high. Conversely, if the model is confident about its prediction for $y$, model parameters will not vary from observing $y$:

$$\mathcal{M}(x) = \mathcal{H}(P(y|x, \mathcal{D})) - \mathbb{E}_{P(\theta|\mathcal{D})}(\mathcal{H}(P(y|x, \theta))). \tag{3}$$

The second term — the Expected Entropy $\mathcal{E}$ — is a measure of the residual uncertainty, ie. the aleatoric uncertainty:

$$\mathcal{E}(x) = \mathbb{E}_{P(\theta|\mathcal{D})}(\mathcal{H}(P(y|x, \theta))). \tag{4}$$

In high dimensional settings, an exact estimation of these different quantities is not tractable, therefore, several approximations and heuristics have been introduced. The softmax variance, i.e. the variance of predictions across model parameters, has been shown to approximate epistemic uncertainty well in certain settings [19–21].

In the context of sequential data, the inherent structure in the data generating process often introduces strong dependencies between the output dimensions, e.g., the tokens in a generated sentence. In cases where there exist weak correlations between tokens, quantifying the different types of uncertainties above can be made tractable by ignoring these dependencies [22], in which case the predictive entropy for a sequence $y = (y_1, y_2, ..., y_L)$ may be approximated as the sum of token-level predictive entropies over the $L$ tokens:

$$\mathcal{U}(x) = \sum_{l=1}^{L} \mathbb{E}_{P(y|x,\mathcal{D})}[\log P(y_l|x, y_{k<l}, \mathcal{D})] \approx \sum_{l=1}^{L} \mathbb{E}_{P(y_l|x,y_{k<l},\mathcal{D})}[\log P(y_l|x, y_{k<l}, \mathcal{D})]. \quad (5)$$

Unlike the standard expectation definition which integrates over all $y$, here only $y_l$ is integrated over, and we condition on $y_{k<l}$, which are obtained from a sample from $P(y|x,\mathcal{D})$. The process is repeated for several of these samples and an average is finally computed to reduce variance.

While the above has been shown to work well in certain experiments [22], valuable information is being discarded when we ignore dependencies across tokens. These dependencies are likely to be informative in applications such as the ones considered in §5.3.1. Naive Monte Carlo-based approximations are expected to perform poorly in high dimensions since the majority of samples will be in regions with negligible contribution to the sum (Appendix A). In an active learning setting, Kirsch et al. [23] derive an importance sampling-based estimator of total uncertainty to mitigate these issues. In Natural Language Processing, alternative approaches using domain-specific metrics such as the BLEU score have been suggested [24], but these are difficult to extend to other domains.

## 3   Importance sampling estimator

We consider discrete output points $y$ that belong to a high-dimensional structured object space $\mathcal{S}$ (e.g., long sequences, large graphs), of cardinality $|\mathcal{S}|$. An exact estimation of the Mutual Information between outcomes $y$ and model parameters $\theta$ (equation 3) is impractical because the expectation involves a sum over exponentially many possible outcomes for $y \in \mathcal{S}$.

In lieu of the heuristics previously discussed, we obtain a principled approximation to the Mutual Information via Monte Carlo estimation using *importance sampling*, with an adequately chosen importance distribution.

We denote $q(\theta) \approx P(\theta|\mathcal{D})$ the learnt approximation to the posterior, and assume we can approximate expectations over model parameters by sampling $M$ independent samples from $q(\theta)$. We can then re-write the Mutual Information $\mathcal{M}$ in equation 3 as follows:

$$\mathcal{M}(x) \approx - \sum_{s=1}^{|\mathcal{S}|} p_s \log p_s + \frac{1}{M} \sum_{m=1}^{M} \sum_{s=1}^{|\mathcal{S}|} p_{s,m} \log p_{s,m} = \sum_{s=1}^{|\mathcal{S}|} \underbrace{\left[ \frac{1}{M} \sum_{m=1}^{M} p_{s,m} \log p_{s,m} - p_s \log p_s \right]}_{h(y_s)}$$

$$(6)$$

where $p_s$ and $p_{s,m}$ are shorthands, respectively, for $P(y = y_s|x, \mathcal{D})$ — the posterior predictive distribution — and $P(y = y_s|x, \theta = \theta_m)$ — the probability of a given output $y_s \in \mathcal{S}$ given $x$ and a sample $\theta_m$ from the approximate posterior distribution over model parameters $q(\theta)$.

We can then obtain a tractable approximation to equation 6 via importance sampling:

$$\mathcal{M}(x) = \sum_{s=1}^{|\mathcal{S}|} h(y_s) \cdot \frac{1}{\bar{p}(y_s)} \cdot \bar{p}(y_s) = \mathbb{E}_{\bar{p}}\left[ h(y) \cdot \frac{1}{\bar{p}(y)} \right] \approx \frac{1}{N} \sum_{s=1}^{N} \left[ h(\tilde{y}_s) \cdot \frac{1}{\bar{p}(\tilde{y}_s)} \right] \quad (7)$$

with $\tilde{y}_s \sim \bar{p}(.)$, where $\bar{p}$ is the importance distribution.

We choose the importance distribution to be the approximate posterior predictive defined over the outputs. We generate an outcome $\tilde{y}_s$ by first sampling a set of parameters $\tilde{\theta}_0$ from the approximate posterior, and then generating $\tilde{y}_s$ from a model defined by that set of parameters $\tilde{\theta}_0$. This distribution

will sample mostly from regions in $\mathcal{S}$ with high probability under the true posterior predictive for input $x$. This is in contrast to a naive sum over all possible outcomes $y$, many of which will have a negligible contribution to the sum. This gives rise to an estimator of Mutual information (obtained with Algorithm 1) with lower variance than its naive Monte Carlo counterpart (see Appendix A).

---

**Algorithm 1:** Importance sampling estimator of MI

---

   **for** $s = 1$ **to** $N$ **do**
      Sample $\tilde{\theta}_0 \sim q(\theta)$ ; $\tilde{y}_s \sim P(y|x, \theta = \tilde{\theta}_0)$ ;
      **for** $m = 1$ **to** $M$ **do**
         Sample $\tilde{\theta}_m \sim q(\theta)$ ; Compute $p_{s,m} = P(y = \tilde{y}_s|z, \theta = \tilde{\theta}_m)$ ;
      **end for**
      Compute $p_s = \frac{1}{M}\sum_{m=1}^{M} p_{s,m}$; $h_s = \frac{1}{M}\sum_{m=1}^{M}(p_{s,m}\log p_{s,m}) - p_s \log p_s$ ;
   **end for**
   Return $\mathcal{M}(x) = \frac{1}{N}\sum_{s=1}^{N}\left[h_s \cdot \frac{1}{p_s}\right]$

---

## 4   Uncertainty-guided optimization in VAE latent space

**Black box optimization in VAE latent space.** We want to optimize the black-box objective $\mathcal{O}$ over a high dimensional discrete object space $\mathcal{S}$. We train a VAE, with encoder $g$ and decoder $f$, to learn a continuous lower-dimensional embedding of objects in $\mathcal{S}$. The optimization of $\mathcal{O}$ is then performed in latent space and the best candidates are subsequently decoded into the original space. As discussed in § 2.1, this may lead to invalid or unrealistic decodings when the decoder $f$ operates in regions different from the ones seen during training. We propose to detect this regime by quantifying the epistemic uncertainty of the decoder for latent points $z$ (note the notation change compared to § 3 where we used $x$ to denote inputs in the general setting): avoiding regions with high epistemic uncertainty for the decoder will make the overall optimization process more efficient by avoiding invalid decodings. We next discuss how we can leverage the uncertainty of the decoder to guide the optimization process for two approaches commonly used in latent optimization settings.

**Bayesian Optimization with an uncertainty-aware surrogate model or uncertainty censoring.** We first train a surrogate model, e.g., a Gaussian Process [25], to predict $\mathcal{O}(x)$ based on its latent representation $z$. We then perform Bayesian Optimization using an appropriate acquisition function (e.g., Upper Confidence Bound or Expected Improvement heuristic). There are two main ways to incorporate the decoder uncertainty to guide this process. The first approach consists in training the surrogate model on an objective that penalizes points with high uncertainty (e.g., optimizing $\mathcal{O}(x) - \alpha \cdot \mathcal{M}(z)$). Another method is to censor proposal points $z$ that would have a Mutual Information $\mathcal{M}(z)$ above a predefined uncertainty threshold $\mathcal{T}$ (e.g., highest value observed on the training data) at each step of a batch Bayesian Optimization process (see Algorithm 2).

---

**Algorithm 2:** Bayesian Optimization with uncertainty censoring

---

 1: Uncertainty threshold $T$, number of new points to generate $N$.
 2: Sample $M$ points uniformly at random from the train set, with latent tensor $Z$ and property vector $P$.
 3: **for** $i = 1$ **to** $N$ **do**
 4:    Train single task GP on $(Z, P)$ and generate $B$ candidate points $(z_k)_{k\in[\![1,B]\!]}$ with predicted properties $(f_k)_{k\in[\![1,B]\!]}$ by sequentially maximizing the Expected Improvement.
 5:    Compute decoder uncertainty $\mathcal{M}(z_k)$ for $k \in [\![1, B]\!]$.
 6:    **if** $\exists k \in [\![1, B]\!]$ s.t. $\mathcal{M}(z_k) \leqslant T$ **then**
 7:       Set new candidate index $k^* = \arg\max_k(f_k)$ s.t. $\mathcal{M}(z_k) \leqslant T$;
 8:    **else**
 9:       $k^* = \arg\min_k(\mathcal{M}(z_k))$.
10:    **end if**
11:    Decode new candidate $z_{k*}$ and obtain true property $p_{k*}$ of decoded candidate.
12:    $(Z, P) \leftarrow$ Concatenate $(z_{k*}, p_{k*})$ and $(Z, P)$.
13: **end for**

---

**Uncertainty-constrained gradient ascent.** A common architecture design when performing black-box optimization in latent space is to jointly train the VAE with an auxiliary network $h$ (Fig.9) that

predicts the value of the black box objective $\mathcal{O}(x)$ from the encoding $z$ of $x$ in latent space [6, 7, 14]. This construct is particularly useful in constrained optimization settings in which we want to perform a local search in latent space to maximize $\mathcal{O}$ while remaining close to a known input object. The joint training consists of optimizing the sum of the VAE loss (i.e., the ELBO) and the loss from the black-box objective prediction (e.g., MSE loss for a continuous output $\mathcal{O}$) via gradient descent, backpropagating gradients through the entire architecture. To optimize objects under this framework (see Algorithm 3), we start from a set of initial points in latent space $z$ — either a random sample of latent points, or a subset of points $x$ that we encode in the latent space ($z = g(x)$). We then compute the gradient $\nabla_z h$ of the auxiliary network with respect to $z$ and perform gradient ascent $z \leftarrow z + \alpha \cdot \nabla_z h$. We repeat this process a few times until satisfying a stopping criteria (e.g., threshold on predicted values $h(z)$ or after a fixed number of gradient updates). Finally, we decode the latest latent positions to obtain the set of candidates $\tilde{x} = f(z)$ and measure their actual properties $\mathcal{O}(\tilde{x})$. We can further improve the quality of the candidate set by censoring the moves in latent during gradient ascent that would result in a value of uncertainty above a predefined threshold $\mathcal{T}$.

---

**Algorithm 3:** Uncertainty-constrained gradient ascent

1: Uncertainty threshold $T$, number of gradient updates $N$, gradient scale $\alpha$.
2: Sample $M$ points from the train set, with latent tensor $Z = (z_k)_{k \in [\![1, M]\!]}$ and property vector $P$.
3: Compute $\nabla_z h(Z)$, where h is the auxiliary network predicting $P$ from $Z$.
4: **for** $i = 1$ **to** $N$ **do**
5:    **for** $k = 1$ **to** $M$ **do**
6:       **if** $\mathcal{M}(z_k + \alpha \nabla_z h(z_k)) \leqslant T$ **then**
7:          $z_k \leftarrow z_k + \alpha \nabla_z h(z_k)$.
8:       **end if**
9:    **end for**
10: **end for**
11: Decode final positions $Z^*$.
12: Obtain true properties $P^*$ of decoded points.

---

# 5 Experimental results

After describing the common experimental setting across applications, we demonstrate the effectiveness of using the uncertainty of the decoder to guide the optimization for constrained digit generation. Our objective is to illustrate the concepts introduced above in a simple and intuitive example. We then move on to quantify the benefits of our approach in the more complex cases of arithmetic expression approximation and molecular generation for drug design.

**Uncertainty estimators and baselines** Across all experiments, we quantify the Mutual Information between outputs and decoder parameters with both the Importance sampling estimator (IS-MI) described in §3 and based on the token independence approximation (TI-MI) described in §2.2. Sampling from model parameters is achieved via Monte Carlo dropout [26]. We compare optimization results with two baselines: the standard approach that fully ignores decoder uncertainty, and an approach in which we censor proposal points with low probability under the prior distribution (standard normal) of the VAEs in latent space (referred to as 'NLLP').

**Optimization** We perform uncertainty-guided optimization in latent space as per the two approaches described in §4. For Bayesian Optimization, we train a single task Gaussian Process as our surrogate model based on a random subset of training points embedded in latent space and their corresponding black-box objective values. We then perform several iterations of batch Bayesian Optimization using the Expected Improvement heuristic as our acquisition function. At each iteration we select a batch of 20 latent vectors by sequentially maximizing the acquisition function. We select the point with the highest predicted target value for which the decoder uncertainty is below a predefined threshold (e.g., $99^{th}$ percentile of decoder uncertainty values observed on the training set) or the one with lowest uncertainty if no point in the batch is below the threshold. We re-train the surrogate model with the newly generated point at each step. For gradient ascent, we randomly sample points from the training set, embed them in latent space, and use these as our starting positions. We then compute the gradient of the auxiliary property network with respect to latent positions and accept proposal moves along these directions if the decoder uncertainty at the corresponding position in latent is below a predefined threshold (e.g., $99^{th}$ percentile of decoder uncertainty on the training set). For practical

considerations, we suggest to always start with a high value for the threshold to not unnecessarily constrain the optimization, and move to stricter uncertainty constraints if the validity or quality is not high enough for the particular use case considered (see Table 10 for an analysis on the impact of that hyperparameter choice). All optimization experiments reported below are carried out 10 times independently with different random seeds.

## 5.1 Illustrative example in digit generation

**Setup** In this first setting, our objective is to generate *valid* images of the digit 3 that are as *thick* as possible. We train a VAE model generating images of the digit 3 jointly with an auxiliary network predicting their thickness. We use a 'Conv-Deconv'[27] architecture for the VAE and a 3-layer feedforward network for property prediction. The underlying data consist of grayscale images of the digit 3 extracted from the MNIST dataset [28] that we discretize to form tensors of binary values. We use the sum of pixel intensities across a given image as a proxy for the thickness of the corresponding digit. An unconstrained optimization in the latent space would ultimately lead to the generation of *invalid* white 'blobs'. To avoid this failure mode and promote validity of the resulting candidate set, we leverage the uncertainty of the decoder network. In order to assess the validity of objects, we independently train a deep Convolutional Neural Network to classify images of the digit 3 (binary classification).

**Results** Latent points with high decoder uncertainty lead to a higher rate of invalid decoded digits (Appendix B.3), which help avoid the aforementioned failure mode during optimization. For both Bayesian Optimization and gradient ascent, we observe that the decoder uncertainty constraints help ensuring the generated digits remain valid, while preserving the ability of the optimization algorithm to increase the thickness (Fig.2).



| Method | Top 10 avg. ↑ | Validity (%) ↑ |
|---|---|---|
| No uncertainty | $-0.96 \pm 0.04$ | $77\% \pm 0.6\%$ |
| NLLP | $-0.97 \pm 0.05$ | $76\% \pm 0.8\%$ |
| TI-MI | $-0.72 \pm 0.06$ | $96\% \pm 0.5\%$ |
| IS-MI | **-0.70 $\pm$ 0.04** | **98% $\pm$ 0.5%** |

Figure 3: **Arithmetic expressions approximation results** Mutual Information-based constraints during Bayesian optimization in latent space help promoting higher validity % of decodings while increasing black-box objective values. Uncertainty threshold values used for censoring candidate points are based on decoder uncertainty values observed on the training data ($95^{th}$ percentile).

Figure 2: **Top 5 decoded digits after optimization** Leveraging decoder uncertainty helps preventing the generation of invalid digits.

## 5.2 Arithmetic expressions approximation

**Setup** We follow an experimental design similar to Kusner et al. [4], in which we seek to optimize univariate arithmetic expressions generated by a formal grammar (rules and examples are provided in Appendix C). The objective is to find an expression that minimizes the mean squared error (MSE) with respect to a predefined target expression $(1/3 * x * sin(x * x))$. More specifically, since the presence of exponentials in expressions may results in very large MSE values, the black-box objective is defined as $-log(1 + MSE)$. We train a 'Character VAE' (CVAE) [4] on $80,000$ expressions generated by the formal grammar, then perform optimization in the latent space.

**Results** We observe that methods leveraging the decoder uncertainty result in almost always valid decodings, and reach higher average values of the black box objective for valid decoded expressions compared to baselines (Fig. 3; gradient ascent results in Appendix C.4). In particular, censoring candidate points based on their probability under the standard normal prior does not help promoting

validity of decodings at all. In this setting with relatively short sequences (arithmetic expressions have at most 19 characters), leveraging the TI-MI or IS-MI estimators leads to comparable performance.

## 5.3 Molecule generation for drug discovery

Molecular generation for drug design seeks to identify new molecules satisfying desired chemical properties. Molecules are typically either represented as sequences of characters, using their SMILES representation [9], or as graphs of atoms [12]. We demonstrate the effectiveness of the approach described in § 4 for these two different representations: experiments with the 'Character VAE' (CVAE) for molecules [6] leverage the SMILES representation, while experiments with the JT-VAE [14] are based on a graph representation of molecules. For both architectures, we trained our models on a set of 250k drug-like molecules from the ZINC dataset [29].

### 5.3.1 Character VAE (CVAE)

**Setup** We jointly train a CVAE model, which learns to encode and decode molecules SMILES strings, along with an auxiliary property network that predicts a target property of these molecules. Following prior work [4, 5, 14], we define the black-box objective as the octanol-water partition coefficient penalized by the synthetic accessibility score and the number of long cycles, (Appendix D.1) and we refer to it as 'Penalized logP' for brevity. Since the SMILES representation of molecules follows a strict syntax that determines whether a given expression is valid or not, we are interested in generating molecules that simultaneously maximize the target property and represent valid SMILES expressions.

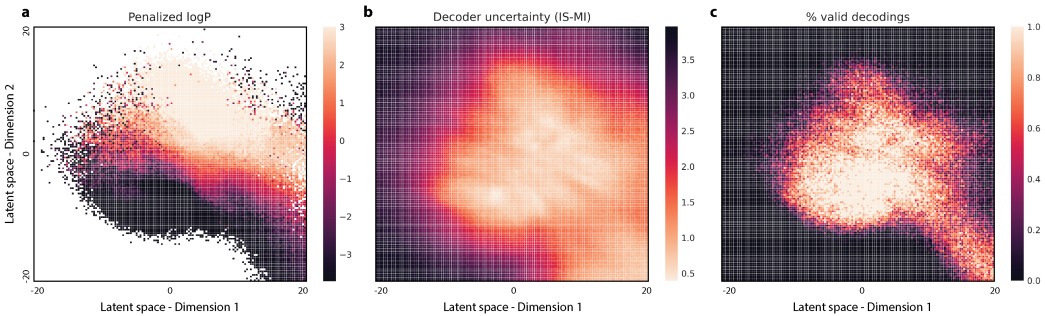

Figure 4: **CVAE latent space visualization.** We apply Principal Component Analysis on the embedding of the full training data and keep the first 2 components. We then create a grid on the resulting 2D-space and measure the penalized logP (a), decoder uncertainty (b) and the proportion of valid decodings in that region (c) (a & c averaged over 300 decodings; b measured via IS by sampling 100 times from the importance distribution, and averaging over 100 samples of model parameters.; for a, white squares correspond to regions where none of the 300 decodings are valid). We observe a strong overlap between decoder uncertainty (b) and validity of decodings (c).

**Results** We first verify that our estimator is able to discriminate points in-distribution (low uncertainty) vs out-of-distribution (high uncertainty). We consider 4 distinct sets of points in latent: embeddings into latent space of a random sample from the train and test sets, random samples from the VAE prior (standard normal) and random samples "far from the prior" (we sample from an isotropic gaussian with standard deviation equal to 10). As can be seen on Fig.5a, uncertainty estimates for the first 3 sets strongly overlap while being disjoint from the estimates corresponding to points far from the prior. Furthermore, we observe a strong correlation between low decoder uncertainty and regions that lead to valid SMILES decodings (Fig. 4). This is corroborated by the analysis described in Fig. 5c: when considering latent points "far from the prior", points for which the decoder uncertainty is lower than a predefined threshold (e.g., maximum value observed on training data) will lead to a significantly higher proportion of valid decoded molecules compared to latent points with uncertainty above the threshold. This is critical as it allows to censor points that will likely lead to invalid decodings, even when we move *far from the prior* in latent space.

For the Bayesian Optimization experiments, we investigate the impact of different bounds on the space we optimize within, as well as different uncertainty thresholds. As we increase the bounds, we typically reach higher optima, at the cost of a higher fraction of invalid decodings during search. We

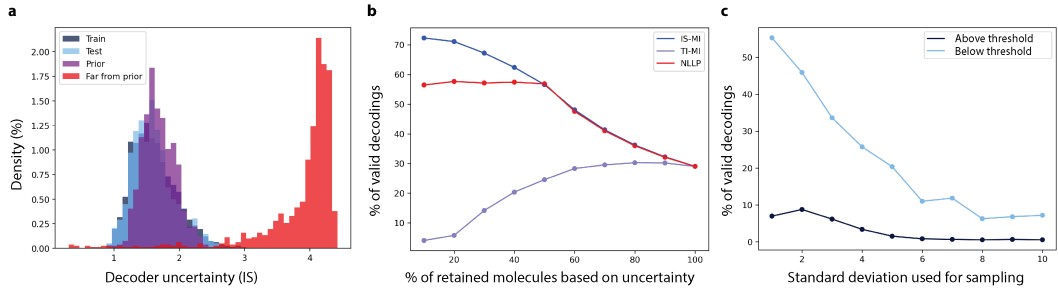

Figure 5: **Uncertainty estimator.** a) Distribution of decoder uncertainty values (IS-MI) for 1k samples for 4 distinct sets (train & test set samples embedded in latent space; samples from the prior; samples far from the prior). b) Valid decodings (%) as a function of the proportion of samples kept based on their uncertainty — eliminating points with high uncertainty first (dataset comprised of 50% samples from test set & 50% of samples far from the prior). The IS-MI estimator has superior ability to identify points leading to invalid decodings. c) Valid decodings (%) for samples from a normal distribution with increasing standard deviation. Samples with decoder uncertainty below a predefined threshold (maximum IS-MI value observed on training data) have a much higher rate of valid decodings. Points above the threshold are very likely to lead to invalid decodings.

Table 1: **CVAE - Bayesian Optimization results.** Censoring proposal points with high decoder uncertainty values with the IS-MI estimator helps increase validity across experiments. As we increase the bounds on the Bayesian Optimization search space, validity % generally decreases but remains 5-10x higher when leveraging IS-MI compared to baselines. This is critical as is helps uncover molecules with very high penalized logP values.

| Search bounds | Decoder uncertainty | Penalized logP | | Validity |
| --- | --- | --- | --- | --- |
| | | Top 1 $\uparrow$ | Avg. top 10 $\uparrow$ | (%) $\uparrow$ |
| 5 | None | $4.0 \pm 0.2$ | $2.5 \pm 0.2$ | $22\% \pm 1.4\%$ |
| | NLLP | $4.2 \pm 0.2$ | $2.7 \pm 0.1$ | $30\% \pm 1.3\%$ |
| | TI-MI | $4.1 \pm 0.3$ | $2.3 \pm 0.1$ | $21\% \pm 0.8\%$ |
| | IS-MI | $\mathbf{4.5 \pm 0.2}$ | $\mathbf{3.0 \pm 0.1}$ | $\mathbf{33\% \pm 1.8\%}$ |
| 10 | None | $3.9 \pm 1.2$ | $-2.3 \pm 2.8$ | $1\% \pm 0.4\%$ |
| | NLLP | $2.9 \pm 0.8$ | $0.5 \pm 0.8$ | $3\% \pm 0.7\%$ |
| | TI-MI | $5.9 \pm 3.6$ | $1.1 \pm 1.5$ | $2\% \pm 0.4\%$ |
| | IS-MI | $\mathbf{6.6 \pm 0.6}$ | $\mathbf{1.6 \pm 0.8}$ | $\mathbf{11\% \pm 0.8\%}$ |
| 15 | None | $10.3 \pm 4.3$ | $5.0 \pm 2.6$ | $1\% \pm 0.3\%$ |
| | NLLP | $3.9 \pm 2.5$ | $0.8 \pm 1.2$ | $1\% \pm 0.3\%$ |
| | TI-MI | $6.7 \pm 3.8$ | $6.4 \pm 3.9$ | $1\% \pm 0.3\%$ |
| | IS-MI | $\mathbf{27.6 \pm 2.2}$ | $\mathbf{9.9 \pm 1.3}$ | $\mathbf{5\% \pm 0.7\%}$ |

obtain higher validity % and penalized logP values when leveraging the decoder uncertainty (Table 1). In this setting, the token-level independence assumption (TI-MI) leads to poor performance compared to the importance sampling-based estimator (IS-MI). Results are also robust to the choice of decoder uncertainty thresholds (D.4).

### 5.3.2 Junction Tree VAE (JT-VAE)

**Setup** We train a Junction Tree VAE model (JT-VAE) [14] using the same dataset of 250k molecules (ZINC) and black-box objective (penalized logP) as for the CVAE experiments. All molecules generated by the JT-VAE are valid by design. However, not all generated molecules will be of high *quality*, which we assess with the quality filters proposed by Brown et al. [30] that aim at ruling out "compounds which are potentially unstable, reactive, laborious to synthesize, or simply unpleasant to the eye of medicinal chemists." We show that it is straightforward to attain state-of-the-art performance in terms of penalized logP values with the basic optimization approaches described in § 4 by moving sufficiently 'far away' in latent, but that in doing so we tend to generate molecules

Table 2: **JT-VAE - Gradient ascent results.** We obtain state-of-the-art performance in terms of penalized logP via gradient ascent. However, most generated molecules are of very low quality (only 1% pass the quality filters from Brown et al. [30]). Leveraging the uncertainty of the decoder (IS-MI) during optimization helps generating molecules with high penalized logP and high quality. NLLP constraints help maintain high quality but lead to suboptimal black-box objective values. Results with different hyperparameters and threshold values for each method are reported in Table 13.

| Decoder uncertainty | Penalized logP - Before filters | | Quality top 10 | Penalized logP - Passing filters | |
| | Top 1 ↑ | Avg. top 10 ↑ | (%) ↑ | Top 1 ↑ | Avg. top 10 ↑ |
| --- | --- | --- | --- | --- | --- |
| None | **23.7 ± 1.3** | **17.0 ± 0.6** | $1\% \pm 1\%$ | $1.2 \pm 1.2$ | $0.3 \pm 0.3$ |
| NLLP | $3.0 \pm 0.1$ | $2.5 \pm 0.1$ | $82\% \pm 6\%$ | $3.0 \pm 0.1$ | $2.0 \pm 0.2$ |
| IS-MI | $8.4 \pm 10.8$ | $6.0 \pm 0.3$ | **89% ± 3%** | **7.7 ± 0.7** | **5.3 ± 0.3** |

that never pass quality filters. Factoring in decoder uncertainty during optimization helps generate new molecules with both high penalized logP values and high quality.

Using notations from § 3, sampling a new object $\tilde{y}_s$ is achieved by successively decoding from the junction tree decoder and then the graph decoder. We then decompose $\log p_{s,m}$ – the log probability of the sampled graph molecule – as the sum of the log probabilities corresponding to the different predictions made by the junction tree decoder and graph decoder, namely the topology and node predictions in the junction tree decoder, and the subgraph prediction in the graph decoder.
We replicate the analysis described in 5.3.1 with the 4 distinct datasets in latent space (i.e., train, test, prior and far from prior) and observe similar results: the histogram of decoder uncertainty values for points "far from the prior" is disjoint from the other three histograms (Appendix E.3), confirming the ability of the estimator to identify out-of-distribution points.

**Results** Both gradient ascent (Table 2) and Bayesian Optimization (Appendix E.4) allow to generate new molecules with state-of-the-art performance in terms of penalized logP (Table 12). However, the majority of these molecules do not pass quality filters. Leveraging decoder uncertainty leads to the generation of high logP and high quality molecules. Using likelihood under the prior (NLLP) to achieve the same is detrimental to optimization performance.

## 6 Discussion and conclusion

**Strengths** Leveraging the uncertainty of the decoder is a *simple* yet *effective* approach to promote the validity or quality of objects generated while optimizing a given black-box property in VAE latent space. It is model architecture-agnostic and does not require model re-training. The only requirement is the ability to sample from decoder parameters – which we have achieved in this work via Monte Carlo dropout given its practical simplicity. In several of the experimental settings, using the decoder uncertainty also helped attain substantially higher values of the black-box objective during optimization (§ 5.3.1 in particular), as the optimization procedure was effectively guided to avoid exploring regions systematically leading to invalid decodings and focusing on more promising regions instead. Lastly, the importance sampling-based estimator introduced in this work is general-purpose and may be relevant to other applications that would benefit from reliable epistemic uncertainty estimates for complex high-dimensional data (e.g., active learning or anomaly detection).

**Limitations** While the algorithm for our estimator is easily parallelizable, it nonetheless results in a computational overhead at each optimization step (which depends on several factors eg., number of samples, decoder architecture, hardware used). In the settings considered in our work (eg., molecular generation), the costs resulting from this overhead are however negligible in comparison to the costs stemming from evaluating the black-box objective (eg., wet lab experiment, expensive simulator).

**Future directions** Developing an approach to backpropagate through the decoder uncertainty estimator may provide compelling alternatives to algorithms presented in § 4. Leveraging the uncertainty of the property network jointly with constraints on the decoder uncertainty may make the search in latent even more robust to invalid decodings. Lastly, our approach may also be used jointly with approaches performing weighted VAE re-training during optimization [17], potentially leading to stronger optimization performance.

## Acknowledgments

P.N. is supported by GSK and the UK Engineering and Physical Sciences Research Council (EPSRC ICASE award no. 18000077). J.M.H.-L. acknowledges support from a Turing AI Fellowship under grant EP/V023756/1. Y.G. holds a Turing AI Fellowship (Phase 1) at the Alan Turing Institute, which is supported by EPSRC grant reference V030302/1.

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
