# Appendix

## A Analysis of variance of uncertainty estimators

We demonstrate the lower variance of the Importance sampling-based estimator compared to the naive Monte Carlo estimator, focusing on the Character VAE for molecular generation setting described in § 5.3.1 and Appendix D.

**Setup.** We sample 1,000 points at random in latent space from an isotropic Gaussian with fixed standard deviation $\sigma$ (we repeat the experiment for different values of the standard deviation). We assess at these points the mutual information between outputs (generated molecule SMILES) and decoder parameters with the Importance sampling-based estimator (IS-MI) described in § 3, and the naive Monte Carlo (MC-MI) equivalent. The MC-MI estimator is obtained directly from equation 6 by sampling output sequences $y_s$ uniformly at random from $\mathcal{S}$, the space of all possible SMILES strings. As per the setting described in Appendix D, we limit the length of molecular sequences to 120 characters from a vocabulary comprised of 34 elements (plus the padding character). Consequently, $\mathcal{S}$ is finite and we can uniformly sample from it by independently sampling characters at each position uniformly at random from the vocabulary. Both estimators are computed by sampling a fixed number of decoder parameters using Monte Carlo dropout [26] (we used 100 model samples in all experiments).

**Results.** We analyze the impact of the number of $y_s$ samples for each estimator on the variance of the corresponding estimators, measured over 10 independent runs. More specifically, we compute the standard deviation over the 10 runs, normalized by the estimator mean across runs. We observe that the IS-MI estimator has a normalized standard deviation 2-10x smaller than the MC-MI estimator across the different experiments (Fig. 6).

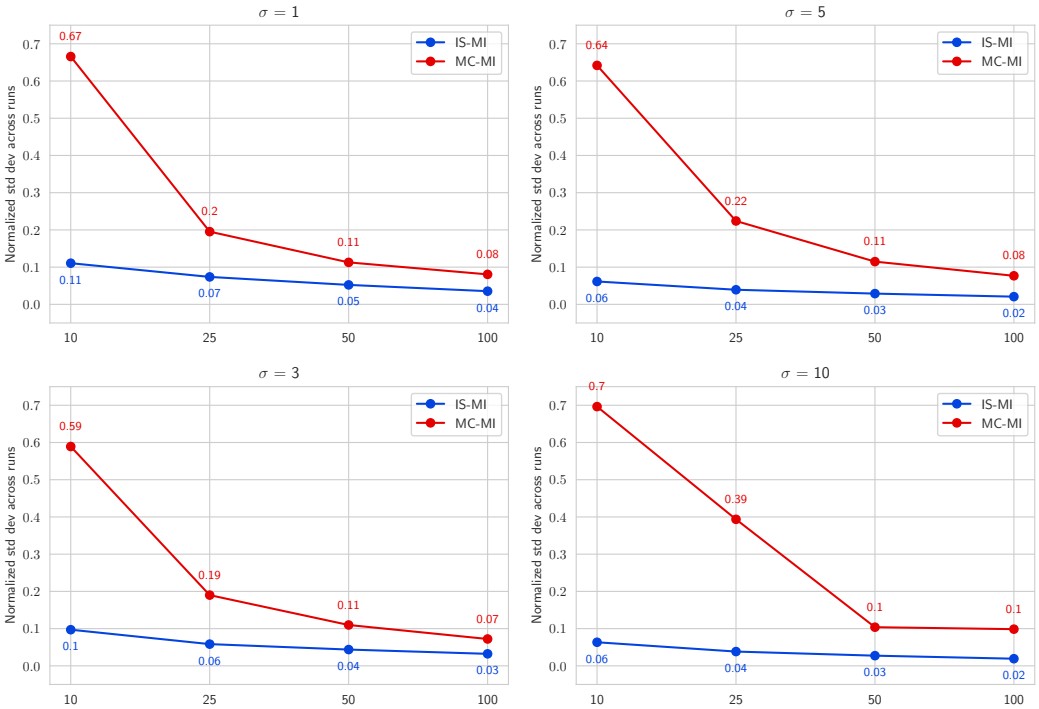

Figure 6: **Variance analysis for uncertainty estimators** Comparison of the normalized standard deviations based on the number of sampled output sequences for the IS-MI Vs MC-MI estimators. We vary the standard deviation of the isotropic Gaussian used to sample points in latent space, from 1 (top left) to 10 (bottom right).

# B  Digit generation experiments

## B.1  Data

**Data source.** The MNIST [28] dataset consists of grayscale images (28x28 pixels) representing handwritten digits. The training dataset is comprised of 60k images and the test dataset is comprised of 10k images.

**Pre-processing.** We use the same train/test split as from the source listed in Table 17, and filter the data to keep images of 3 digits only. We then binarize the data, using a low pixel intensity threshold value ($10^{th}$ percentile of pixel intensity values observed on the training data). No data augmentation is used at train time nor at inference. We use the sum of all pixel intensities in the image as a proxy of the thickness of the digits, which provides strong empirical results.

## B.2  Model details

**Architecture.** We jointly train a variational autoencoder with an auxiliary network (the "Property network") predicting digit thickness based on latent representation (see Fig. 9). For the VAE, we use a "Conv-Deconv" architecture[27]. All model parameters are summarized in table 3.

Table 3: **Digit generation - Model architecture details**

| Component | Description |
|---|---|
| **Encoder** | • 4 consecutive convolutional layers with 28, 56, 56 and 224 filters respectively, kernel sizes 4,4,3 and 4 respectively, with batch norm and RELU activations after each convolutional layer
• Continuous latent space of dimension 2 |
| **Decoder** | • 4 2D-transposed convolutional layers with 224, 56, 56 and 28 filters respectively, kernel sizes 4,3,4 and 4 respectively, with dropout 0.2 and RELU activations after each convolutional layer |
| **Property network** | • 3-layer feedforward network with 100 units each
• RELU activations (after each layer except the final one) and dropout 0.1 |

We assess the validity of generated digits with a separately trained Convolutional Neural Network (CNN) that learns to classify images of 3 digits (binary classification). This network is comprised of two convolution layers (with 23 and 64 filters respectively), followed by a max pool layer and two fully-connected layers (with 9,216 and 128 units respectively). RELU activations are used throughout, except for the final layer where a sigmoid activation is used for the binary classification.

**Training.** We train the joint architecture by minimizing the sum of the VAE ELBO and the mean squared error (MSE) on the thickness prediction task. We use the Adam algorithm [31] with a learning rate of $10^{-3}$, batch size of $512$, weight decay $10^{-5}$ for 300 epochs. We anneal the KL divergence with a sigmoid schedule for the first 30 epochs to avoid potential posterior collapse.

The independent CNN classifier network is trained by minimizing the binary cross entropy loss. We use the Adam optimizer, with learning rate $10^{-4}$, batch size 512 and train for 120 epochs.

## B.3  Uncertainty estimator

We compute the Importance sampling-based estimator following Algorithm 1, with 100 samples from model parameters and 100 binary images $y$ sampled from the importance distribution. We use Monte Carlo dropout [26] to sample decoder parameters, , given its simplicity of implementation and since it is a well established method for uncertainty quantification Abdar et al. [32].

**Uncertainty histograms.** We evaluate the IS-MI estimator values in different regions of latent space. We consider 4 distinct sets of points:

- **'Train':** Embeddings into latent space of 5k images sampled randomly from the train set;
- **'Test':** Embeddings into latent space of 5k images sampled randomly from the test set;
- **'Prior':** 5k random samples from the VAE prior;
- **'Far from prior':** 5k random samples from an isotropic Gaussian with standard deviation equal to 20.

We observe an almost perfect overlap between the distributions of decoder uncertainty values for the 'Train' and 'Test' sets, and both have a strong overlap with the 'Prior' set (Fig. 7). These first 3 sets are all fairly disjoint

from the last set ('far from prior'), confirming the ability of the uncertainty estimators to properly identify 'out-of-distribution' points.

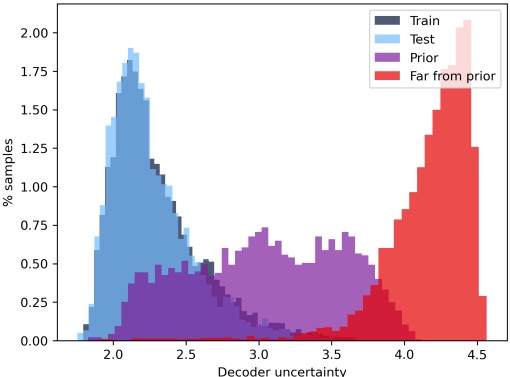

Figure 7: **Digit generation - Decoder Uncertainty distribution** Distribution of decoder uncertainty values based on the IS-MI estimator, evaluated on the 4 distinct datasets defined in Appendix B.3

## B.4 Optimization details

We follow the gradient ascent and Bayesian Optimization approaches described at the beginning of § 5. We provide below all hyperparameter values used for each method.

**Gradient ascent.** We start from 300 point sampled at random from the training set and then embedded in latent space. We perform 10 gradient update steps with a value of alpha equal to 50 (using notations from § 4). When censoring points based on the IS-MI estimator we used a conservative decoder uncertainty threshold, defined as the $75^{th}$ percentile of the IS-MI estimator values observed on training data.

**Bayesian optimization.** The single task Gaussian Process is initially trained on 1,000 images sampled at random from the training set that we then embed in latent space. We use the Expected Improvement as our acquisition function and sequentially generate 10 new digits overall (re-training the Gaussian Process after each acquisition). When leveraging the IS-MI estimator to guide the optimization, we censor proposal points based on the same uncertainty threshold as for gradient ascent ($75^{th}$ percentile of the IS-MI estimator values on training data).

Results for the gradient ascent and Bayesian Optimization experiments are presented on Figure 2. Using the uncertainty of the decoder during gradient ascent of Bayesian Optimization helps maximizing thickness of the generated digits while preventing the them to be invalid white 'blobs'.

## C  Arithmetic expression experiments

### C.1  Data

**Data source.** We use the same dataset as in Kusner et al. [4] which consists of univariate arithmetic expressions that are randomly generated from the following grammar:

$$
\begin{aligned}
&\text{S} \ \rightarrow \ \text{S '+' T | S '*' T | S '/' T | T} \\
&\text{T} \ \rightarrow \ \text{'(' S ')' | 'sin(' S ')' | 'exp(' S ')'} \\
&\text{T} \ \rightarrow \ \text{'x' | '1' | '2' | '3'}
\end{aligned}
$$

where S and T are non-terminals and the symbol '|' separates the possible production rules generated from each non-terminal. For instance, the following univariate arithmetic expressions can be generated from this grammar: $\sin(2)$, $x/(3+1)$, $2+x+\sin(1/2)$, and $x/2 * \exp(x)/\exp(2*x)$.

**Target.** The objective of the optimization in this setting is to find an expression that minimizes the mean squared error (MSE) with respect to a predefined target expression: $1/3 * x * \sin(x * x)$. Specifically, we measure the MSE between the target and proposal expressions over 1000 input values $x$ that are linearly-spaced between $-10$ and $10$. Since the presence of exponentials in expressions may results in very large MSE values, the black-box objective to maximize is actually defined as $-\log(1 + MSE)$.

## C.2 Model details

**Architecture.** We jointly train a variational autoencoder and a property network which predicts the target defined above. For the VAE, we use an architecture identical to the CVAE in Kusner et al. [4], with a convolutional neural network (CNN) encoder and a Recurrent Neural Network (RNN) decoder. All model parameters are summarized in table 4.

Table 4: **Arithmetic expressions approximation - Model architecture details**

| Component | Description |
| --- | --- |
| **Encoder** | • 3 consecutive 1D convolutional layers with 2,3 and 4 filters respectively (kernel size 5), with batch norm and RELU activations after each convolutional layer 
 • Continuous latent space of dimension 25 |
| **Decoder** | • A stack of 3 Gated recurrent unit (GRU) layers [33] with hidden dimension 100, dropout 0.2 (between layers) and RELU activations except for the last layer which has a softmax activation over the arithmetic expressions vocabulary |
| **Property network** | • 3-layer feedforward network with 200 units each 
 • RELU activations (after each layer except the final one) and dropout 0.2 |

**Training.** We train the joint architecture by minimizing the sum of the VAE ELBO and $\log(1 + MSE)$ between an expression and the target one (as per Appendix C.1). We minimize the loss with the Adam algorithm [31] with a learning rate of $10^{-3}$ and batch size of 600 for 80 epochs. We annealed the KL divergence with a sigmoid schedule for the first 10 epochs to avoid potential posterior collapse.

## C.3 Uncertainty estimators

We replicate an analysis analogous to the one described in Appendix B.3, sampling 1k points in latent space for each of the 4 datasets. The IS-MI estimator is computed with 100 samples from model parameters via Monte Carlo dropout and 100 expressions samples from the importance distribution. Similarly to what we observed before, the distribution of decoder uncertainty values for the 'Train', 'Test' and 'Prior' sets have a strong overlap and are fairly disjoint from the 'Far from prior' distribution (Fig. 8).

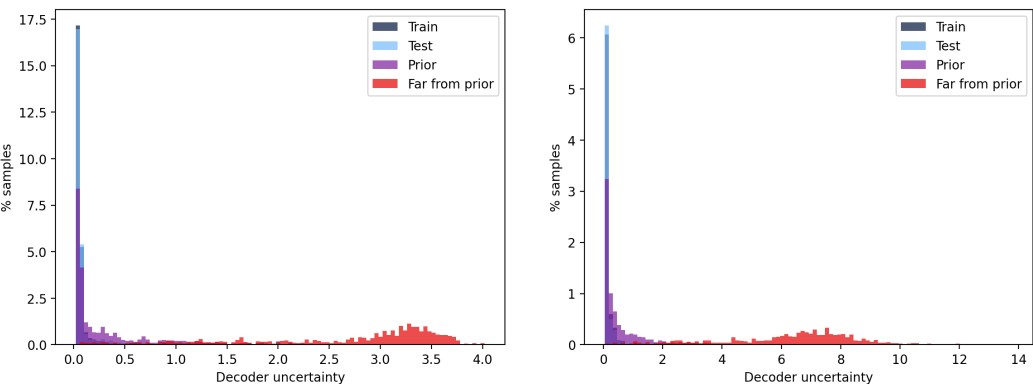

Figure 8: **Arithmetic expressions approximation - Decoder Uncertainty distribution**. Analysis for the IS-MI estimator (left) and the TI-MI estimator (right). Both provide relatively good separation of 'out-of-distribution' points in this setting.

## C.4 Detailed optimization results

We follow the gradient ascent and Bayesian Optimization approaches described at the beginning of § 5. We provide below all hyperparameter values used in each method.

**Gradient ascent.** We start from 500 point sampled at random from the training set and then embedded in latent space. We perform 10 gradient update steps with a value of alpha equal to 10 (using notations from § 4). When censoring points based on the IS-MI estimator we used a decoder uncertainty threshold defined as the $95^{th}$ percentile of the IS-MI estimator values observed on training data. Detailed results in Table 5.

Table 5: **Arithmetic expression - Gradient ascent results.**

| Decoder uncertainty | - log(1+MSE) | | | | Validity (%) ↑ |
| --- | --- | --- | --- | --- | --- |
| | Top 1 ↑ | Top 2 ↑ | Top 3 ↑ | Avg. top 10 ↑ | |
| None | $-0.32 \pm 0.04$ | $-0.38 \pm 0.02$ | $-0.42 \pm 0.01$ | $-0.46 \pm 0.02$ | $60\% \pm 0.6\%$ |
| NLLP | $-0.22 \pm 0.05$ | $-0.37 \pm 0.02$ | $-0.39 \pm 0.01$ | $-0.4 \pm 0.01$ | $63\% \pm 0.8\%$ |
| TI-MI | $-0.17 \pm 0.05$ | **-0.31 $\pm$ 0.05** | $-0.35 \pm 0.03$ | **-0.36 $\pm$ 0.01** | **99% $\pm$ 0.1%** |
| IS-MI | **-0.13 $\pm$ 0.04** | $-0.32 \pm 0.03$ | **-0.33 $\pm$ 0.04** | **-0.36 $\pm$ 0.01** | $98\% \pm 0.2\%$ |

**Bayesian optimization.** The single task Gaussian Process is initially trained on 500 expressions sampled at random from the training set that we then embed in latent space. We sequentially generate 250 new arithmetic expressions (re-training the Gaussian Process after each acquisition). When using the IS-MI estimator to guide the optimization, we censor proposal points based on the same uncertainty threshold as for gradient ascent (ie., $95^{th}$ percentile of the IS-MI estimator values observed on training data). Detailed results in Table 6.

Table 6: **Arithmetic expression - Bayesian Optimization results.**

| Decoder uncertainty | - log(1+MSE) | | | | Validity (%) ↑ |
| --- | --- | --- | --- | --- | --- |
| | Top 1 ↑ | Top 2 ↑ | Top 3 ↑ | Avg. top 10 ↑ | |
| None. | $-0.57 \pm 0.06$ | $-0.69 \pm 0.05$ | $-0.78 \pm 0.05$ | $-0.96 \pm 0.04$ | $77\% \pm 0.6\%$ |
| NLLP | $-0.61 \pm 0.03$ | $-0.75 \pm 0.05$ | $-0.81 \pm 0.05$ | $-0.97 \pm 0.05$ | $76\% \pm 0.8\%$ |
| TI-MI | **-0.40 $\pm$ 0.07** | $-0.55 \pm 0.03$ | $-0.64 \pm 0.05$ | $-0.72 \pm 0.06$ | $96\% \pm 0.5\%$ |
| IS-MI | $-0.41 \pm 0.05$ | **-0.52 $\pm$ 0.05** | **-0.59 $\pm$ 0.05** | **-0.70 $\pm$ 0.05** | **98% $\pm$ 0.5%** |

In both the gradient ascent and Bayesian Optimization experiments, we obtain higher values of the black-box objective as well as a higher proportion of valid generated expressions (nearing 100% in both cases) when leveraging decoder uncertainty. We get comparable results with the IS-MI and TI-MI estimators in this setting with relatively short sequences.

## D  Molecule generation experiments with CVAE

### D.1  Data

**Data source.** We use a dataset of 250k drug-like molecules from the ZINC database [29]. Each molecule is represented via its SMILES representation [9], ie. as a sequence of characters (from a vocabulary of 34 elements, plus the padding character). Following [6], molecule length is capped at 120, and shorter strings are space-padded to this length.

**Target.** The black-box objective in this set of experiments is the 'penalized logP', defined as the octanol-water partition coefficient penalized by the synthetic accessibility score and the number of long cycles. We follow prior work [4, 5, 14, 11] and compute this metric as follows:

$$\text{Penalized log P}(x) = \widehat{\log P(x)} - \widehat{SAS(x)} - \widehat{cycle(x)} \tag{8}$$

where $\log P(x)$ is the octanol-water partition coefficient, $SAS(x)$ is the synthetic accessibility score, $cycle(x)$ counts the number of rings that have more than six atoms, and the $\hat{}$ operator represents the standard normalization based on the raw training subset from ZINC (ie. subtracting the mean of the training set, and dividing by the standard deviation).

### D.2  Model details

**Architecture.** We adopt a model architecture similar to Gómez-Bombarelli et al. [6]: the encoder is comprised of 3 convolutional layers, the decoder is composed of a stack of 3 GRU layers [33] and the property network has a simple feed forward architecture with 3 hidden layers. A detailed description is provided in Table 7.

**Training.** The total loss we minimize is the sum of the VAE ELBO and the MSE loss on the black-box property prediction task. We train the network with the Adam algorithm [31] with a learning rate of $5.10^{-4}$ (reduced by a factor 2 with a patience of 10 epochs) for 150 epochs total. We anneal the KL divergence with a sigmoid

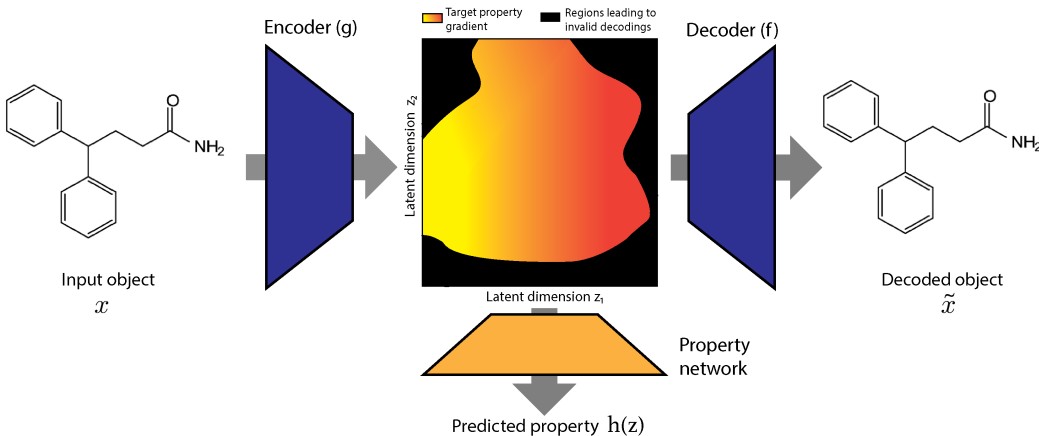

Figure 9: **Joint training architecture** A Variational autoencoder (VAE) is jointly trained with an auxiliary network predicting the value of the black-box objective from the latent space encoding. Optimization is then carried out in latent space via gradient ascent or Bayesian optimization

Table 7: **Molecular generation with CVAE - Model architecture details**

| Component | Description |
|---|---|
| **Encoder** | • 3 consecutive 1D convolutional layers with 9,9 and 10 filters respectively, kernel sizes 9,9 and 11 respectively, with batch norm and RELU activations after each convolutional layer
• Continuous latent space of dimension 56 |
| **Decoder** | • A stack of 3 Gated recurrent unit (GRU) layers [33] with hidden dimension 500, with dropout 0.2 (between layers) and RELU activations except for the last layer which has a softmax activation over the SMILES vocabulary.
• At each step, the character generated at the previous step is concatenated with the latent embedding and fed as input |
| **Property network** | • 3-layer feedforward network with 1,000 units each
• RELU activations (after each layer except the final one) and dropout 0.2 |

schedule for the first 30 epochs to avoid potential posterior collapse. We also use teacher forcing on the character sampled at each time step in the decoder during training, and gradient clipping (upper bound set to 10) to avoid exploding gradients.

### D.3 Uncertainty estimators

Similar to the previous two experimental settings, we first assess our uncertainty estimator by examining the distribution of its values on the 4 datasets defined in Appendix B.3, using 1k samples for each dataset. We compute the IS-MI estimator based on Algorithm 1, use Monte Carlo dropout sampling to obtain 100 samples from decoder parameters, and sample 100 molecule SMILES from the importance distribution (Fig. 10). In this setting, the TI-MI estimator provides very poor uncertainty estimates as it assigns very low uncertainty values for a majority of 'out-of-distribution' points that were sampled far from the prior (Fig.10). This is consistent with what we see in Fig.5b. In this experiment, we analyze the proportion of valid decodings when keeping the x% points we are most certain about, based on the various estimators considered (ie. IS-MI, NLLP and TI-MI). If low uncertainty for a given estimator corresponds to high validity, then the % of valid decodings should increase as we keep a narrower set of most confident points. This is what we observe for the IS-MI estimator, unlike the TI-MI estimator which incorrectly selects points leading to invalid decodings as the lowest uncertainty points.

### D.4 Detailed optimization results

**Gradient ascent.** We start from 200 point sampled at random from the training set and embedded in latent space. We perform 10 gradient update steps with a value of alpha equal to 20. In the experiments where we impose a

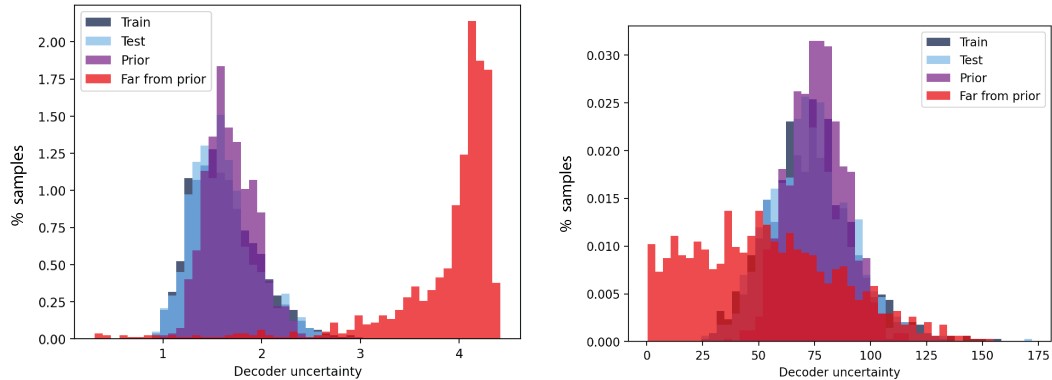

Figure 10: **Molecular generation with CVAE - Decoder uncertainty distribution**. Analysis for the IS-MI estimator (left) and the TI-MI estimator (right)

maximum threshold on the decoder uncertainty, we set that threshold as the $99^{th}$ percentile of corresponding estimator values observed on the training data. Leveraging the IS-MI estimator during optimization helps reaching higher values of the black-box objective (Table 8). While constraints imposed with the NLLP baselines do help maintaining a higher % of valid decodings, using this baseline is detrimental to optimization performance.

Table 8: **Molecular generation with CVAE - Gradient ascent results.**

| Decoder uncertainty | Penalized logP | | | | Validity (%) ↑ |
|---|---|---|---|---|---|
| | Top 1 ↑ | Top 2 ↑ | Top 3 ↑ | Avg. top 10 ↑ | |
| None | $5.1 \pm 0.2$ | $4.6 \pm 0.2$ | $4.1 \pm 0.1$ | $1.6 \pm 0.5$ | $4.1\% \pm 0.4\%$ |
| NLLP | $4.8 \pm 0.1$ | $4.6 \pm 0$ | $4.4 \pm 0$ | $4.3 \pm 0$ | **56.6% $\pm$ 0.9%** |
| TI-MI | $3.3 \pm 2$ | $4.6 \pm 0.1$ | $4.4 \pm 0.1$ | $0.8 \pm 1.8$ | $4.1\% \pm 0.5\%$ |
| IS-MI | **5.8 $\pm$ 0.2** | **5.4 $\pm$ 0.1** | **5.1 $\pm$ 0.1** | **4.9 $\pm$ 0.1** | $21.5\% \pm 1\%$ |

**Bayesian Optimization.** We train a single task Gaussian Process (GP) on 500 points sampled at random from the training set and embedded in latent. We use the Expected Improvement as our acquisition function, sequentially generate 100 new molecules (and re-train the GP after each acquisition). Similar to our gradient ascent experiments, we set the uncertainty threshold to the $99^{th}$ percentile of corresponding estimator values observed on the training data. We observe that when using the IS-MI estimator, we not only increase the % of valid decodings by 1.5-10x compared to baselines, but we also reach higher values of the 'penalized logP' objective across all settings (Table. 9). These results are robust to the choice of distribution decile used to define the uncertainty threshold, as shown in Table.10.

# E Molecule generation experiments with JTVAE

## E.1 Data

We refer the reader to Appendix D.1 as we used the same experimental setting as for the CVAE experiments, ie. we train our model on a subset of 250k drug-like molecules from the ZINC database, and seek to optimize the 'penalized logP' metric.

## E.2 Model architecture

**Architecture.** We jointly train a Junction Tree VAE model (JT-VAE) with a property network predicting the 'penalized logP' property based on latent representation, leveraging the same architecture design as in Jin et al. [14] for the constrained optimization task. The only difference we introduce is the incorporation of dropout layers in the junction tree decoder and graph decoder to allow sampling from the decoder parameters via Monte Carlo dropout (see Table 11).

**Training.** In line with the experiments discussed above, and following the same training procedure as per Jin et al. [14], we minimize the sum of the VAE loss and the MSE on the black-box prediction task.

Table 9: **Molecular generation with CVAE - Bayesian Optimization results.** NA values in the table corresponds to no valid molecule decoded across the 10 independent runs. 'Validity' measures the proportion of generated molecules that correspond to valid SMILES expressions. 'Unicity' corresponds to the ratio of the number of distinct generated molecules to the total number of generated molecules. 'Novelty' is defined as the proportion of generated molecules that were not present in the training data.

| Search bounds | Decoder uncertainty | Penalized logP | | | | Validity (%) ↑ | Unicity (%) ↑ | Novelty (%) ↑ |
|---|---|---|---|---|---|---|---|---|
| | | Top 1 ↑ | Top 2 ↑ | Top 3 ↑ | Avg. top 10 ↑ | | | |
| 5 | None | $4.0 \pm 0.2$ | $3.5 \pm 0.2$ | $3.2 \pm 0.2$ | $2.5 \pm 0.2$ | $21.9\% \pm 1.4\%$ | $100\% \pm 0\%$ | $100\% \pm 0\%$ |
| | NLLP | $4.2 \pm 0.2$ | $3.6 \pm 0.1$ | $3.2 \pm 0.1$ | $2.7 \pm 0.1$ | $29.6\% \pm 1.3\%$ | $100\% \pm 0\%$ | $100\% \pm 0\%$ |
| | TI-MI | $4.1 \pm 0.2$ | $3.5 \pm 0.2$ | $3.1 \pm 0.1$ | $2.3 \pm 0.1$ | $21.0\% \pm 0.8\%$ | $100\% \pm 0\%$ | $100\% \pm 0\%$ |
| | IS-MI | $\mathbf{4.5 \pm 0.2}$ | $\mathbf{3.7 \pm 0.2}$ | $\mathbf{3.5 \pm 0.2}$ | $\mathbf{3.0 \pm 0.1}$ | $\mathbf{33.2}\% \pm \mathbf{1.8}\%$ | $100\% \pm 0\%$ | $100\% \pm 0\%$ |
| 10 | None | $3.9 \pm 1.1$ | $-1.9 \pm 6.9$ | $-14.4 \pm 4$ | $-2.3 \pm 2.8$ | $1.1\% \pm 0.4\%$ | $80\% \pm 8\%$ | $80\% \pm 8\%$ |
| | NLLP | $2.9 \pm 0.7$ | $0.2 \pm 1.4$ | $2.3 \pm 0.8$ | $0.5 \pm 0.8$ | $2.8\% \pm 0.7\%$ | $60\% \pm 16\%$ | $60\% \pm 16\%$ |
| | TI-MI | $5.9 \pm 3.3$ | $-1.9 \pm 1.7$ | $0.2 \pm 0.7$ | $1.1 \pm 1.5$ | $1.6\% \pm 0.4\%$ | $80\% \pm 13\%$ | $80\% \pm 13\%$ |
| | IS-MI | $\mathbf{6.6 \pm 0.5}$ | $\mathbf{4.6 \pm 0.6}$ | $\mathbf{3.6 \pm 0.3}$ | $\mathbf{1.6 \pm 0.8}$ | $\mathbf{10.6}\% \pm \mathbf{0.8}\%$ | $\mathbf{99}\% \pm \mathbf{1}\%$ | $\mathbf{100}\% \pm \mathbf{0}\%$ |
| 15 | None | $10.3 \pm 3.9$ | $-3.0 \pm 2.7$ | NA | $5.0 \pm 2.6$ | $1.0\% \pm 0.3\%$ | $80\% \pm 11\%$ | $80\% \pm 11\%$ |
| | NLLP | $3.9 \pm 2.3$ | $-4.6 \pm 4.9$ | NA | $0.8 \pm 1.2$ | $1.0\% \pm 0.3\%$ | $64\% \pm 16\%$ | $64\% \pm 16\%$ |
| | TI-MI | $6.7 \pm 3.6$ | $0.0 \pm 1.7$ | NA | $6.4 \pm 3.9$ | $1.1\% \pm 0.3\%$ | $73\% \pm 15\%$ | $73\% \pm 15\%$ |
| | IS-MI | $\mathbf{27.6 \pm 2.1}$ | $\mathbf{15.4 \pm 3.9}$ | $\mathbf{7.8 \pm 3.2}$ | $\mathbf{9.9 \pm 1.3}$ | $\mathbf{5.5}\% \pm \mathbf{0.7}\%$ | $\mathbf{96}\% \pm \mathbf{2}\%$ | $\mathbf{100}\% \pm \mathbf{0}\%$ |

Table 10: **Molecular generation with CVAE - Impact of the choice of decoder uncertainty threshold on Bayesian Optimization results.** This analysis focuses on the IS-MI estimator only. NA values in the table corresponds to no valid molecule decoded. The second column represents the percentile of the IS-MI estimator values on the training data used to define the threshold for censored Bayesian Optimization.

| Search bounds | Uncertainty threshold | Penalized logP | | | | Validity (%) ↑ |
|---|---|---|---|---|---|---|
| | | Top 1 ↑ | Top 2 ↑ | Top 3 ↑ | Avg. top 10 ↑ | |
| 5 | None | $4.0 \pm 0.2$ | $3.5 \pm 0.2$ | $3.2 \pm 0.2$ | $2.5 \pm 0.2$ | $21.9\% \pm 1.4\%$ |
| | Median | $4.2 \pm 0.1$ | $\mathbf{3.9 \pm 0.1}$ | $\mathbf{3.7 \pm 0.1}$ | $\mathbf{3.4 \pm 0.1}$ | $\mathbf{54.8}\% \pm \mathbf{1.2}\%$ |
| | P90 | $4.3 \pm 0.1$ | $3.8 \pm 0.1$ | $3.6 \pm 0.1$ | $3.3 \pm 0.1$ | $48.2\% \pm 2.3\%$ |
| | P95 | $4.3 \pm 0.1$ | $3.8 \pm 0.2$ | $3.6 \pm 0.1$ | $3.2 \pm 0.1$ | $44.0\% \pm 1.7\%$ |
| | P99 | $\mathbf{4.5 \pm 0.2}$ | $3.7 \pm 0.2$ | $3.5 \pm 0.2$ | $3 \pm 0.1$ | $33.2\% \pm 1.8\%$ |
| | Max | $4.3 \pm 0.2$ | $3.6 \pm 0.2$ | $3.1 \pm 0.2$ | $2.6 \pm 0.2$ | $26.9\% \pm 1.8\%$ |
| 10 | None | $3.9 \pm 1.2$ | $-12.7 \pm 6.9$ | $-14.4 \pm 4$ | $-2.3 \pm 2.8$ | $1.1\% \pm 0.4\%$ |
| | Median | $\mathbf{7.6 \pm 0.7}$ | $4.7 \pm 0.5$ | $3.8 \pm 0.3$ | $2.5 \pm 0.4$ | $\mathbf{12.4}\% \pm \mathbf{0.7}\%$ |
| | P90 | $\mathbf{7.6 \pm 0.7}$ | $\mathbf{4.8 \pm 0.5}$ | $3.6 \pm 0.3$ | $2.5 \pm 0.4$ | $11.1\% \pm 0.7\%$ |
| | P95 | $7.0 \pm 0.9$ | $4.5 \pm 0.6$ | $\mathbf{3.9 \pm 0.4}$ | $\mathbf{2.6 \pm 0.4}$ | $12.0\% \pm 0.7\%$ |
| | P99 | $6.6 \pm 0.6$ | $4.6 \pm 0.6$ | $3.6 \pm 0.3$ | $1.6 \pm 0.8$ | $10.6\% \pm 0.8\%$ |
| | Max | $5.7 \pm 0.8$ | $3.8 \pm 0.3$ | $2.9 \pm 0.3$ | $0.9 \pm 0.7$ | $9.1\% \pm 0.8\%$ |
| 15 | None | $10.3 \pm 4.3$ | $-3.0 \pm 2.7$ | NA | $5.0 \pm 2.6$ | $1.0\% \pm 0.3\%$ |
| | Median | $21.6 \pm 3.9$ | $10.3 \pm 4.2$ | $8.3 \pm 3.8$ | $6.1 \pm 1.9$ | $5.8\% \pm 0.8\%$ |
| | P90 | $24.0 \pm 3.3$ | $14.1 \pm 4.2$ | $7.8 \pm 3.4$ | $7.6 \pm 1.7$ | $5.8\% \pm 0.7\%$ |
| | P95 | $26.5 \pm 2.5$ | $\mathbf{17.2 \pm 4.1}$ | $\mathbf{8.9 \pm 4.1}$ | $8.4 \pm 1.6$ | $\mathbf{6.2}\% \pm \mathbf{0.7}\%$ |
| | P99 | $\mathbf{27.6 \pm 2.2}$ | $15.4 \pm 3.9$ | $7.8 \pm 3.2$ | $\mathbf{9.9 \pm 1.3}$ | $5.5\% \pm 0.7\%$ |
| | Max | $23.7 \pm 3.3$ | $13.6 \pm 3.7$ | $7.1 \pm 3.8$ | $8.2 \pm 2.2$ | $5.5\% \pm 0.7\%$ |

### E.3 Uncertainty estimators

We compute the IS-MI estimator based on Algorithm 1. Following notations from Algorithm 1, we sample a molecule $\tilde{y}_s$ by successively decoding from the junction tree decoder and then the graph decoder. We then decompose $\log p_{s,m}$ as the sum of the log probabilities corresponding to each prediction made by the junction tree decoder and graph decoder (for the graph outcome generated in the previous step), namely the topology and node predictions in the junction tree decoder, and the subgraph prediction in the graph decoder. Similar to the setting discussed in Appendix B.3, we inspect the distribution of the IS-MI values obtained on the same 4 datasets (using 1k samples for each dataset), and estimate the mutual information with 100 samples from decoder parameters and 100 molecules sampled from the importance sampling distribution. We observe similar results as before: overlap between IS-MI distributions on the 'Train', 'Test' and 'Prior' sets. This overlap is

Table 11: **Molecular generation with JT-VAE - Model architecture details.** All components are identical to the JT-VAE architecture used for constrained optimization from Jin et al. [14], except for the dropout layers detailed below

| Component | Description |
|---|---|
| **Encoder** | • Junction tree encoder and molecular graph encoder
• Continuous latent space of dimension 56 |
| **Decoder** | • Junction tree decoder with dropout layer (0.2 drop rate) applied to the input and the output of the GRU used for message passing, and dropout layer (0.2 rate) applied right before the final layer for both the topology prediction and node prediction networks
• Molecular graph decoder with dropout layer (0.2 rate) applied right before the final layer of the subgraph prediction network |
| **Property network** | • 2-layer feedforward network with 450 units and 1 unit resp., $\tanh$ activation for the first layer, no activation for the second, and dropout 0.2 before both layer |

stronger between the first two sets as the embedding of the training data in latent does not necessarily follow a standard normal distribution after model training. The distribution of IS-MI values on the 'Far from prior' set is disjoint from the first 3 sets, with the highest values obtained on this set.

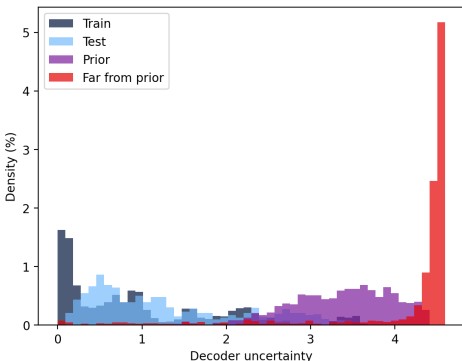

Figure 11: **Molecular generation with JT-VAE - Decoder uncertainty distribution.**

### E.4    Detailed optimization results

All molecules generated by the JT-VAE are valid by design. However, not all generated molecules will be of high *quality*, as measured for example by the quality filters from Brown et al. [30] discussed in § 5.3.2. Since in this molecular generation setting, our objective is to generate new molecules with high penalized logP values that could be used as potential drugs, we want to ensure that the candidate drugs we shortlist for further investigation also pass these quality filters. In each optimization experiment described below, we prioritize a small number of candidate molecules (eg., top 10 molecules with highest logP values) and use the quality filters from Brown et al. [30] as a proxy for the subsequent (costly) verification of these candidates by medicinal chemists. If a generated molecule does not pass these quality filters, its penalized logP value is assigned to a default value (eg., average penalized logP value on the training set). In all optimization experiments, we estimate mutual information with 100 samples from decoder parameters, and a single molecule sampled from the importance distribution.

**Gradient ascent.** We start from 100 molecules sampled at random from the training set, that we then embed in latent space. We perform 100 gradient update steps with a large value of $\alpha$ (as per notations in § 4), eg., 100 or 200. This leads state-of-the-art performance in terms of penalized logP values (see Table 12 and Fig. 12). However, as we move 'further away' in latent space, the quality of generated molecules tends to degrade. By setting an upper bound on the uncertainty of the decoder (eg., $95^{th}$ percentile of IS-MI values observed on the training data) during optimization, we are able to generate molecules with both high penalized logP values and high quality (see Table 13). Selecting different uncertainty threshold values enable to reach different trade-offs between quality and black-box objective. A similar approach with upper bounds in terms of NLLP values (eg., threshold defined as $95^{th}$ or $99^{th}$ percentiles of NLLP values on the training data) does help promoting high quality molecules but leads to much lower penalized logP values.

Table 12: **Molecular generation - Top optimization performance.** We achieve state-of-the-art performance on the molecular generation task using a JT-VAE model jointly trained with an auxiliary network predicting penalized logP from latent embeddings (as per §3.3 of [14]) and performing gradient ascent as described in § 4. Results were obtained by embedding in latent space 100 points selected at random from the test set and then performing 100 gradient updates with $\alpha = 200$. We report mean performance over 10 runs, as well as the best generated molecules across these 10 runs.

| Model | Optimization method | Penalized logP | | |
| --- | --- | --- | --- | --- |
| | | Top 1 ↑ | Top 2 ↑ | Top 3 ↑ |
| JT-VAE [14] | Bayesian Optimization | 5.30 | 4.93 | 4.49 |
| MolDQN [15] | Reinforcement learning | 11.84 | 11.84 | 11.82 |
| GraphAF [16] | Reinforcement learning | 12.23 | 11.29 | 11.05 |
| CCGF [11] | Chance-constrained optimization | 12.32 | 11.79 | 11.61 |
| ChemBO [34] | Bayesian Optimization | 18.39 | - | - |
| JT-VAE [17] | Bayesian Optim. & retraining (median of 5 runs) | 21.20 | 15.34 | 15.34 |
| JT-VAE [17] | Bayesian Optim. & retraining (best over 5 runs) | 27.84 | 27.59 | 27.21 |
| JT-VAE (ours) | Gradient ascent (mean of 10 runs) | 23.65 | 21.17 | 19.45 |
| JT-VAE (ours) | Gradient ascent (best over 10 runs) | **30.81** | **30.00** | **29.82** |

Table 13: **Molecular generation with JT-VAE - Gradient ascent results.** The 'Uncertainty threshold' column represents the percentile of training values used to define the uncertainty threshold.

| $\alpha = 100$ | | | | | | | | | | |
| --- | --- | --- | --- | --- | --- | --- | --- | --- | --- | --- |
| Decoder uncertainty | Uncertainty threshold | Penalized logP - Before filters | | | | Quality top 10 (%) ↑ | Penalized logP - Passing filters | | | |
| | | Top 1 ↑ | Top 2 ↑ | Top 3 ↑ | Avg. top 10 ↑ | | Top 1 ↑ | Top 2 ↑ | Top 3 ↑ | Avg. top 10 ↑ |
| None | None | **22.4 ± 0.9** | **19.9 ± 0.6** | **18.7 ± 0.4** | **16.6 ± 0.3** | 3% ± 2% | 4.3 ± 2.2 | 0.0 ± 0.0 | 0.0 ± 0.0 | 0.4 ± 0.2 |
| NLLP | P95 | 3.4 ± 0.1 | 2.9 ± 0.1 | 2.6 ± 0.1 | 2.4 ± 0.1 | 71% ± 4% | 3.3 ± 0.2 | 2.7 ± 0.1 | 2.5 ± 0.1 | 1.8 ± 0.1 |
| | P99 | 3.8 ± 0.1 | 3.4 ± 0.1 | 3.2 ± 0.1 | 3.0 ± 0.1 | **89% ± 3%** | 3.8 ± 0.1 | 3.3 ± 0.1 | 3.2 ± 0.1 | 2.7 ± 0.1 |
| | Max | 4.5 ± 0.1 | 4.2 ± 0.1 | 4.0 ± 0.1 | 3.8 ± 0.1 | 82% ± 4% | 4.4 ± 0.1 | 4.0 ± 0.1 | 3.8 ± 0.1 | 3.1 ± 0.1 |
| IS-MI | P95 | 11.4 ± 1.4 | 8.1 ± 0.3 | 7.7 ± 0.2 | 7.6 ± 0.2 | 81% ± 5% | 8.3 ± 0.3 | 7.7 ± 0.2 | **7.4 ± 0.2** | **5.8 ± 0.3** |
| | P99 | 17.5 ± 1.4 | 13.3 ± 0.9 | 11.4 ± 0.7 | 10.3 ± 0.5 | 48% ± 6% | **9.9 ± 0.7** | **8.4 ± 0.2** | 6.9 ± 0.8 | 3.9 ± 0.4 |
| | Max | 19.4 ± 1.1 | 16.7 ± 1.0 | 15.1 ± 0.9 | 13.0 ± 0.6 | 19% ± 4% | **9.9 ± 1.1** | 4.9 ± 1.5 | 2.6 ± 1.2 | 1.8 ± 0.3 |
| $\alpha = 200$ | | | | | | | | | | |
| Decoder uncertainty | Uncertainty threshold | Penalized logP - Before filters | | | | Quality top 10 (%) ↑ | Penalized logP - Passing filters | | | |
| | | Top 1 ↑ | Top 2 ↑ | Top 3 ↑ | Avg. top 10 ↑ | | Top 1 ↑ | Top 2 ↑ | Top 3 ↑ | Avg. top 10 ↑ |
| None | None | **23.7 ± 1.3** | **21.2 ± 0.8** | **19.5 ± 0.8** | **17.0 ± 0.6** | 1% ± 1% | 1.2 ± 1.2 | 0.0 ± 0.0 | 0.0 ± 0.0 | 0.1 ± 0.1 |
| NLLP | P95 | 3.0 ± 0.1 | 2.8 ± 0.1 | 2.7 ± 0.1 | 2.5 ± 0.05 | 82% ± 6% | 3.0 ± 0.1 | 2.7 ± 0.1 | 2.6 ± 0.1 | 2.0 ± 0.2 |
| | P99 | 3.3 ± 0.1 | 3.0 ± 0.1 | 2.8 ± 0.1 | 2.6 ± 0.1 | 80% ± 4% | 3.2 ± 0.1 | 2.9 ± 0.1 | 2.7 ± 0.1 | 2.1 ± 0.1 |
| | Max | 3.6 ± 0.1 | 3.2 ± 0.1 | 3.0 ± 0.1 | 2.7 ± 0.1 | 81% ± 4% | 3.6 ± 0.1 | 3.1 ± 0.1 | 3.0 ± 0.1 | 2.2 ± 0.1 |
| IS-MI | P95 | 8.4 ± 0.8 | 6.8 ± 0.4 | 6.4 ± 0.4 | 6.0 ± 0.3 | **89% ± 3%** | 7.7 ± 0.7 | 6.5 ± 0.3 | 6.1 ± 0.3 | **5.3 ± 0.3** |
| | P99 | 15.3 ± 1.6 | 9.5 ± 1.0 | 7.2 ± 0.3 | 7.1 ± 0.4 | 69% ± 4% | 7.6 ± 0.3 | **6.8 ± 0.3** | **6.3 ± 0.3** | 4.0 ± 0.1 |
| | Max | 20.5 ± 1.3 | 17.3 ± 1.4 | 13.8 ± 1.2 | 11.4 ± 0.8 | 31% ± 7% | **8.6 ± 1.3** | 6.1 ± 1.1 | 3.4 ± 1.1 | 2.3 ± 0.5 |

**Bayesian Optimization.** We train a single task Gaussian Process (GP) on 500 points sampled at random from the training set and embedded in latent. We use the Expected Improvement as our acquisition function, sequentially generate 500 new molecules (and re-train the GP after each acquisition). In experiments in which we impose an upper bound on decoder uncertainty, we set that bound as the $99^{th}$ percentile of decoder uncertainty values observed on the training data. Similar to what we observe in the gradient ascent experiments, leveraging the IS-MI estimator helps generating candidate molecules with both high 'penalized logP' values and high quality (Table 14). The NLLP of proposal points at each step of the batch Bayesian Optimization tend to always be above the NLLP threshold ($99^{th}$ percentile of values on the training data). In these situations, we select the point with lowest NLLP value from the proposal batch as described in § 5. This explains why the results obtained with NLLP are closer to what we obtain without any constraint during optimization, and why we lose the ability to increase the quality of generated molecules in this setting.

# F Additional discussion

**What we thought would work, but did not work.** In § 4 we presented two possible approaches for incorporating the decoder uncertainty within Bayesian Optimization in latent space: the uncertainty censoring approach and the uncertainty-aware surrogate model approach. While we have used the former extensively across experiments in § 5, the latter has not delivered strong results. A possible interpretation would be that the latter approach

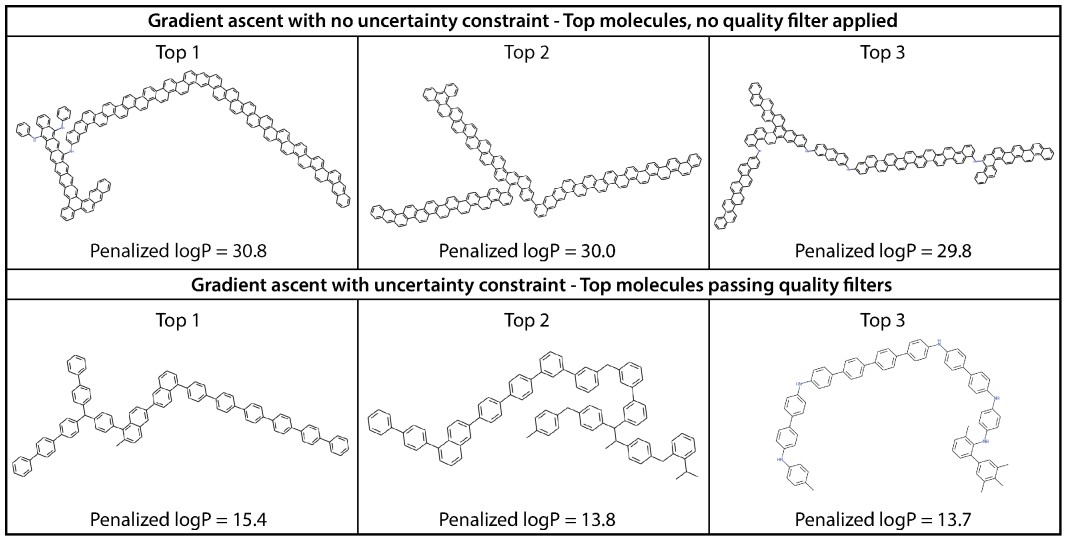

Figure 12: **Top molecules generated via gradient ascent with a JT-VAE** ($\alpha = 200$).

Table 14: **Molecular generation with JT-VAE - Bayesian Optimization results.**

| Decoder uncertainty | Penalized logP - Before filters | | | | Quality top 10 (%) ↑ | Penalized logP - Passing filters | | | |
|---|---|---|---|---|---|---|---|---|---|
| | Top 1 ↑ | Top 2 ↑ | Top 3 ↑ | Avg. top 10 ↑ | | Top 1 ↑ | Top 2 ↑ | Top 3 ↑ | Avg. top 10 ↑ |
| None | **18.6 ± 1.5** | **14.1 ± 1.8** | **11.5 ± 1.2** | **9.6 ± 0.7** | 18% ± 6% | 7.1 ± 2.2 | 3.0 ± 1.1 | 2.2 ± 1.1 | 1.4 ± 0.4 |
| NLLP | 15.7 ± 1.5 | 12.2 ± 0.8 | 9.7 ± 0.5 | 8.7 ± 0.4 | 23% ± 5% | 7.8 ± 0.6 | 4.6 ± 1.0 | 1.1 ± 0.7 | 1.5 ± 0.3 |
| IS-MI | 14.3 ± 1.7 | 9.2 ± 1.0 | 7.6 ± 0.9 | 6.3 ± 0.4 | **47% ± 3%** | **9.7 ± 2.1** | **4.8 ± 0.4** | **4.2 ± 0.3** | **2.1 ± 0.3** |

may heavily penalize optimization directions for which the gradient of the black-box objective is aligned with the gradient of decoder uncertainty, while the former would allow moving along these directions up until the uncertainty threshold is exceeded.

**Societal impact.** We introduce a general approach to improve the black-box optimization of complex high-dimensional discrete objects in VAE latent space. The potential applications of this field are very broad – from more effective drug or protein design to improved automatic program synthesis. The estimator of decoder uncertainty we have introduced may also be leveraged in a wide range of other areas that would benefit from reliable uncertainty estimates in high dimensional settings (eg., active learning, outlier detection).

# G   Reproducibility

**Code and dependencies.** Our codebase is available at the following address: `https://github.com/pascalnotin/uncertainty_guided_optimization`. It was fully developed in Pytorch (v1.7) [35]. All Bayesian Optimization experiments were conducted with the BoTorch package [36] (available under MIT license). For molecule generation experiments, we used the rdkit (https://github.com/rdkit/rdkit) and guacamol packages (https://github.com/BenevolentAI/guacamol), available under a BSD 3-Clause and a MIT license respectively. In addition, we have made use of the code repositories listed in Table 15 when developing our approach and conducting experiments.

Table 15: **Code repositories used**

| Setting | Data source |
|---|---|
| Grammar VAE | `https://github.com/mkusner/grammarVAE` |
| JT-VAE (official repo) | `https://github.com/wengong-jin/icml18-jtnn` |
| JT-VAE (Python 3 implementation) | `https://github.com/Bibyutatsu/FastJTNNpy3` |
| Quality filters | `https://github.com/PatWalters/rd_filters` |

**Compute resources.** All experiments were carried out with a single GPU (Titan RTX). We summarize compute usage for the different experiments in Table 16.

Table 16: **Compute usage per experiment summary**

| Experiment | Avg. compute time per iteration (GPU hrs) | | | |
| --- | --- | --- | --- | --- |
| | **Digit generation** | **Arithmetic expression** | **CVAE** | **JTVAE** |
| Model training | <0.5 | 0.5 | 12 | 50 |
| Decoder uncertainty histograms | 2 | 2 | 3 | 40 |
| Bayesian optimization | <0.5 | <0.5 | 2 | 14 |
| Gradient ascent | <0.5 | <0.5 | 1 | 5 |

**Data sources.** We list the raw data sources used across all experiments in Table 17: the MNIST dataset (Creative Commons Attribution-Share Alike 3.0 license), the arithmetic expressions dataset from Kusner et al. [4], and the ZINC data (see also `https://zinc.docking.org/`) used in Gómez-Bombarelli et al. [6] (Apache License 2.0).

Table 17: **Data sources summary**

| Setting | Data source |
| --- | --- |
| Digit generation | `http://yann.lecun.com/exdb/mnist/` |
| Arithmetic expression | `https://github.com/mkusner/grammarVAE/tree/master/data` |
| Molecular generation | `https://github.com/aspuru-guzik-group/chemical_vae/tree/master/models/zinc` |