# OpenReview forum: "Improving black-box optimization in VAE latent space using decoder uncertainty"
_NeurIPS.cc/2021/Conference — NeurIPS 2021 Poster_

### Official Review · Reviewer_4uN8 · 2021-07-14

**Rating:** 6
**Confidence:** 4

**Summary:**

The paper considers the problem of black-box optimization of high-dimensional discrete objects where the inputs are embedded into a continuous latent space (typically using a deep generative model) and the optimization process takes place in this latent space. The main hypothesis in the paper is that existing methods for this problem might not be robust because they explore uncertain parts of the latent space with no training data in that region. The paper proposes to leverage the epistemic uncertainty of the decoder (measured in terms of the mutual information between model parameters and outputs) in order to navigate the latent space for black-box optimization. The mutual information is computed via an importance sampling estimate and employed in Bayesian optimization (BO) and gradient based search on the latent space. Experiments are performed on digit generation, arithmetic expression and molecular generation benchmarks.


**Limitations And Societal Impact:**

Yes, Societal impact is discussed in the appendix.

**Main Review:**

- One nice thing about the paper is that the proposed approach is complementary to existing methods and can be employed in a drop-in replacement.


- Although the importance sampling estimator gives principled uncertainty estimate, it is an extra computational overhead which can be substantial. The estimator requires sampling from the model posterior (accomplished in the paper via MC-dropout) which is already a very challenging and active research problem. Please discuss the implications of this challenging step in computing the estimator. Please provide a computational complexity analysis of using the proposed approach on top of existing methods. Some empirical analysis in terms of raw wall-clock time would be very useful.


- The paper proposes two modifications to BO over latent space: censor points above an uncertainty threshold or training GP surrogate model by penalizing points with high uncertainty. However, in BO, the surrogate model's uncertainty is essential in exploring the function. How does the proposed method interact with the uncertainty of the surrogate model. It seems like it might push BO to over-exploit if the uncertainty threshold is not tuned properly.


- Please provide a discussion on the criterion for selecting the right uncertainty threshold and how robust the proposed approach is to this parameter.


- The experimental comparison in the paper is slightly weak because the baselines are a bit naive. The right method to compare for this setting is Tripp et al. [1] which also proposes a drop-in replacement approach for latent-space optimization. Please add some discussion discussing the pros and cons of the proposed approach with [1].


- A minor point but the references section can be improved significantly. Many references are missing even conference names. Please fix that.

References

[1] Tripp, A., Daxberger, E., & Hernández-Lobato, J. M. (2020). Sample-efficient optimization in the latent space of deep generative models via weighted retraining. NeurIPS, 2020

**Time Spent Reviewing:**

6

---

> ### Author Response · Authors · 2021-08-10
> **Authors' response to Reviewer 4uN8**
>
> Dear reviewer,
>
> We sincerely thank you for your time and valuable feedback. We kindly ask that you read our comment above, addressed to all reviewers, as well as our responses to your specific comments. Below we address your specific comments as best as we can, and we hope you will engage with us actively during the discussion period to clarify any remaining points.
>
> **[Although the importance sampling estimator gives principled uncertainty estimate, it is an extra computational overhead which can be substantial. The estimator requires sampling from the model posterior (accomplished in the paper via MC-dropout) which is already a very challenging and active research problem. Please discuss the implications of this challenging step in computing the estimator. Please provide a computational complexity analysis of using the proposed approach on top of existing methods. Some empirical analysis in terms of raw wall-clock time would be very useful.]**
>
> Please see section F in supplementary material for a discussion on this point. In a nutshell, in the settings considered in our work, computational costs/timing are dwarfed by the costs needed to evaluate the black-box objectives (e.g., costly wet lab experiment) and therefore not a primary concern in practice. Additionally, the importance sampling-based MI estimation is parallelizable, which significantly reduces the computational overhead if implemented properly. In practice, the overhead is going to be driven by several factors (e.g., number of samples for IS-MI evaluation, decoder architecture, hardware used). Regarding the sampling from the model posterior, we used MC dropout given its simplicity of implementation and since it is a well established method for uncertainty quantification (see Abdar et al. “A Review of Uncertainty Quantification in Deep Learning: Techniques, Applications and Challenges”, 2020; Gawlikowski et al. “A Survey of Uncertainty in Deep Neural Networks”, 2021). Based on your feedback and input from other reviewers, we realize the points covered in section F are important and thus suggest moving this section right before the conclusion in the main manuscript.
>
> **[The paper proposes two modifications to BO over latent space: censor points above an uncertainty threshold or training GP surrogate model by penalizing points with high uncertainty. However, in BO, the surrogate model's uncertainty is essential in exploring the function. How does the proposed method interact with the uncertainty of the surrogate model. It seems like it might push BO to over-exploit if the uncertainty threshold is not tuned properly.]**
>
> While the uncertainty of the decoder does not necessarily correlate with the uncertainty of the surrogate model, you do have a point in general. In the uncertainty censoring approach, proposal points depend only on the acquisition function (hence on the surrogate's model uncertainty only) and the decoder uncertainty approach is then used solely to select the point with highest predicted objective value that is below the uncertainty threshold. Consequently it does not impact the relative exploration-exploitation tradeoff of the surrogate model, and does not seem to impact our algorithm's performance in experiments. In the other variant with the uncertainty-aware surrogate, the additional penalty based on uncertainty of the decoder may lead to a different exploration-exploitation tradeoff (e.g., if the decoder uncertainty is positively correlated with the surrogate model uncertainty). This may explain some of the less compelling results obtained with this particular variant (see also the discussion in section F of supplementary material).
>
> **[Please provide a discussion on the criterion for selecting the right uncertainty threshold and how robust the proposed approach is to this parameter.]**
>
> At a high level, a stricter uncertainty threshold will result in higher validity/quality at the cost of potentially lower black-box objective values for the decoded objects. Please see table 10 in supplementary material where we look at the impact of this hyperparameter on optimization performance. For practical considerations, we suggest to always start with a high value for the threshold (e.g., 99-th percentile) to not necessarily constrain the optimization, and move to stricter uncertainty constraints if the validity / quality is not high enough for the particular use case considered. We will add this guidance on line 210 in the manuscript.
>
> **[The experimental comparison in the paper is slightly weak because the baselines are a bit naive. The right method to compare for this setting is Tripp et al. [1] which also proposes a drop-in replacement approach for latent-space optimization. Please add some discussion discussing the pros and cons of the proposed approach with [1].]**
>
> We see our method and the one introduced in Tripp et al. as complementary methods which could be combined together to further improve optimization performance. The method introduced in Tripp et al successively re-trains the VAE weighing more heavily observations with high black-box objective values: the objective is to learn a representation that decodes more broadly to points with good objective values. However, this latent representation could still present unreliable decoding in some regions. Conversely, our approach takes a fixed VAE as input (no re-training needed; the latent embedding manifold is fixed) and will instead seek to guide the optimization in latent space to regions where decoding is more reliable. The two methods could actually be combined to obtain improved results. Based on your feedback, we have also included a head to head comparison of the two methods in the additional experiments achieving SOTA performance on the molecular generation task (see point A in our response to all reviewers above).
>
> **[A minor point but the references section can be improved significantly. Many references are missing even conference names. Please fix that]**
>
> This will be fixed.
>
> We hope that our response and the new experiments have adequately addressed your concerns. We would greatly appreciate it if you could engage with us during the discussion period on any remaining barriers to raising your score.
>
> Thank you,
> The Authors

---

> > ### Comment · Reviewer_4uN8 · 2021-08-29
> > **Thank you for clarification**
> >
> > Thank you for your reply to my comments. This is a nice contribution and I will be happy with the paper getting accepted.

---

### Official Review · Reviewer_nbh7 · 2021-07-16

**Rating:** 6
**Confidence:** 3

**Summary:**

This paper propose to improve the latent space optimization (LSO) framework through incorporating uncertainty of decoders. The current  latent sace optimization algorithms are based on (a) training an generative autoencoder of candidate solutions for the optimization problem and (b) searching over latent codes that corresponds to high objective function. Searching over latent codes that are unseen during training may lead to the decoder generating invalid outputs. The authors propose to constrain the search process based on (newly proposed) importance sampling-based estimation of decoder uncertainty. Experiments validate the useful-ness of the approach compared to the two baselines: naive LSO and LSO constrained by prior distribution of latent codes.

**Ethical Concerns:**

There are no ethical issues with this paper.

**Limitations And Societal Impact:**

The authors adequately addressed the limitations and potential negative societal impact of their work.

**Main Review:**

Overall, I believe this paper propose a sensible approach to tackle an important problem. It seems clear that incorporating decoder uncertainty is beneficial to the latent space optimization framework. However, I believe additional experiments should be conducted to evaluate the useful-ness of the proposed importance sampling scheme to estimate decoder uncertainty.

Pros:
- This paper tackels an important problem of the latent space optimization framework that has gained interest.
- The proposed approach is simple and sensible.
- Experiments are conducted over various domains to demonstrate its general usage.

Cons:
- My main concern is on the experiments missing baselines for measuring the decoder uncertainty. There are many works that propose algorithms for evaluating uncertainty of (sequential) generative model and detecting out-of-distribution samples. For example, the authors can compare to the algorithms listed in [Ren et al., 2019].
-  The authors should cite & discuss their position of their paper with respect to [Griffith et al., 2020] who performs a constrained Bayesian optimization for LSO.
- In the supplemenent, it is mentioned that the authors use Monte Carlo dropout to estimate posterior distribution of the neural network parameters. I think this is an imporatant detail that should be shown in the main paper.
- (minor) The authors could provide results on various score functions for the molecular optimization task. There are many scoring functions that are more relavent to the real applications like drug discovery. For example, the authors can consider the GuacaMol benchmark [Brown et al., 2019] or GSK & JNK [Li et al., 2019]

Question:
- I was able to understand that decoder uncertainty helps to generate latent codes that are decoded in a valid & realistic way (ratio of valid & high-quality molecules). However, I am curious what is the explanation for how improving the decoder uncertainty leads to better performance of LSO.

[Ren et al., 2019] Likelihood Ratios for Out-of-Distribution Detection
[Griffith et al., 2020] Constrained Bayesian optimization for automatic chemical design using variational autoencoders
[Brown et al., 2019] GuacaMol: Benchmarking Models for De Novo Molecular Design
[Li et al., 2019] Multi-objective de novo drug design with conditionalgraph generative model

**Time Spent Reviewing:**

4

---

> ### Author Response · Authors · 2021-08-10
> **Authors' response to Reviewer nbh7**
>
> Dear reviewer,
>
> We sincerely thank you for your time and valuable feedback. We kindly ask that you read our comment above, addressed to all reviewers, as well as our responses to your specific comments. Below we address your specific comments as best as we can, and we hope you will engage with us actively during the discussion period to clarify any remaining points.
>
> **[My main concern is on the experiments missing baselines for measuring the decoder uncertainty. There are many works that propose algorithms for evaluating uncertainty of (sequential) generative model and detecting out-of-distribution samples. For example, the authors can compare to the algorithms listed in [Ren et al., 2019].]**
>
> We do benchmark our importance sampling-based MI estimator with the negative log likelihood under the prior baseline (a simple baseline available out-of-the-box), as well as the method introduced by Malinin and Gales ("Uncertainty estimation in autoregressive structured prediction", ICLR 2021), which is to our knowledge the most relevant (and recent) method to compare against in the settings we focus on in this work (high-dimensional structured datasets). Could you please clarify if you think there are more appropriate baselines for us to compare against?
> To our knowledge there is no other existing baseline to measure uncertainty in these high dimensional settings (as noted by Malinin & Gales). Please note regarding the work from Ren et al that:
> * Authors define OOD points as follows: “In this paper, we consider an input (x,y) to be OOD if y∈Y: that is, the class y does not belong to one of the K in-distribution classes.” (first paragraph of Background section). This definition does not apply to our context (we do not focus on classification)
> * The metrics 1-8 (on page 5) do not apply to our setting (all depend on access to a class y)
> * Metric 9 does not apply to VAE settings (requires explicit modeling of the likelihood) and require access to an ensemble of models
>
> **[The authors should cite & discuss their position of their paper with respect to [Griffith et al., 2020] who performs a constrained Bayesian optimization for LSO.]**
>
> We will include a reference to [Griffith et al., 2020] and describe this work in the related work section. The approach introduced in [Griffith et al., 2020] requires some feedback (in the form of whether a constraint is satisfied or not) to learn what regions of the latent space are likely to produce reasonable decodings. In the case of molecules represented as text strings, this feedback is whether the decoded text string is a valid molecule. However, in many practical applications, this feedback information is not readily available. For example, the molecules decoded by the Junction Tree VAE are always valid but some may be unrealistic and should be rejected. Our method can do this because it does not require any feedback information, it only uses an estimate of the decoding uncertainty.
>
> **[In the supplemenent, it is mentioned that the authors use Monte Carlo dropout to estimate posterior distribution of the neural network parameters. I think this is an imporatant detail that should be shown in the main paper.]**
>
> We do mention that Monte Carlo dropout is used in the MI estimation in the main text (lines 193-194), but will make that more prominent.
>
> **[(minor) The authors could provide results on various score functions for the molecular optimization task. There are many scoring functions that are more relavent to the real applications like drug discovery. For example, the authors can consider the GuacaMol benchmark [Brown et al., 2019] or GSK & JNK [Li et al., 2019] ]**
>
> Following your suggestions, we are providing additional metrics from the Guacamol benchmark (validity, uniqueness, novelty) for our two molecular generation settings (see point B in our response to all reviewers above). Please note that since the main contributions of this work are domain-agnostic, we focused experiments on the most frequently used metrics in the existing literature for each setting -- this is the case for the penalized logP metric, used in the large majority of ML papers on molecular generation. As several papers have pointed out (e.g,. Zhou et al in "Optimization of molecules with deep reinforcement learning", 2019), this metric -- while popular -- has its limitations. The other metrics in the Guacamol benchmark [Brown et al., 2019] or GSK & JNK [Li et al., 2019] may be helpful complements to address these gaps for molecular generation.
>
> **[I was able to understand that decoder uncertainty helps to generate latent codes that are decoded in a valid & realistic way (ratio of valid & high-quality molecules). However, I am curious what is the explanation for how improving the decoder uncertainty leads to better performance of LSO.]**
>
> We observe this phenomenon in the Character VAE experiments. In that setting, the validity of points sampled at random far from the prior drops close to 0% (see Fig 5c). Unfortunately, the points with the highest black-box objective values are also far from the prior. Therefore, a successful optimization method is one able to find these high-value points needles in that haystack of invalid points. We observe that leveraging the uncertainty of the decoder achieves just that.
>
> We hope that our response and the new experiments have adequately addressed your concerns. We would greatly appreciate it if you could engage with us during the discussion period on any remaining barriers to raising your score.
>
> Thank you,
> The Authors

---

> > ### Comment · Reviewer_nbh7 · 2021-08-10
> > **Thank you for the clarification.**
> >
> > Thank you for the detailed response. I acknowledge my misunderstandings in the baselines. I have raised the score and lowered down the confidence of my readings accordingly.

---

> > > ### Author Response · Authors · 2021-08-20
> > > **Post rebuttal response**
> > >
> > > Dear reviewer,
> > >
> > > Thank you very much for your response and feedback during reviews.
> > >
> > > Kind regards,
> > > The authors

---

### Official Review · Reviewer_VYPR · 2021-07-16

**Rating:** 7
**Confidence:** 3

**Summary:**

The author(s) note that modelling discrete objects with VAEs that use a continuous latent representation provides an easy way to optimize an auxiliary "design" function, which maps discrete data-space objects to a scalar fitness score, via indirectly optimizing the latent space by back propagating "design" function gradients through the VAEs decoder. In other words, this method circumvents discrete optimization methods in favor of simpler continuous optimization strategies. The author(s) correctly observe this can be problematic when such an optimization results in values that have low density under the variational aggregate posterior (i.e. latent codes that produce invalid objects since the decoder has never seen them before). The author(s) propose an importance-sampling-based estimate to measure decoder uncertainty as a way to discourage auxiliary "design" function optimization from converging on latent codes that decode into invalid objects. The author(s) integrate this uncertainty into both gradient ascent and Bayesian optimization regimes to experimentally showcase their proposals.

**Limitations And Societal Impact:**

The author(s) do not really address these issues. Can the author(s) please think of times their uncertainty estimation might produce false estimates or when their optimization proposals might avoid less probably but valid regions of latent space that might be associated with a less-represented group?

**Main Review:**

### Originality:
- I am unfamiliar with the latest and greatest mutual information estimation techniques. However, I do know for certain there is a large body of related research on MI estimation. Could the author(s) please discuss which aspects of their approach are novel? From lines 103-106, it sounds the decomposition (equations 2-4) is not novel. The paper reads as if importance-sampling these quantities is novel. Can the author(s) please confirm this is the case? I think the paper would benefit on a relevant works related to MI estimation alone, since this is a major contribution of the paper.

### Quality:
- The author(s) mention a variety of related works the first paragraph of section 2.1. Yet, many of these methods do not appear as experimental baselines in section 5. Their absence makes it difficult to assess the proposals of this paper. In particular, how is a researcher to select between existing methods or the ones proposed?

### Clarity:
- While each section reads clearly on its own, I was unable to comprehensively understand the implementation of the manuscript's proposals. This difficulty is the principal reason for my low score. I hope the author(s) can address the following issues.
- Conventional VAE lexicon uses x for data space z for latent space. Section 2.2 uses x for inputs and y for discrete outputs. Wouldn't the inputs be y as well since a VAE reconstructs the input from itself? Or is y reconstructed from x? How do x and y in section 2.2 relate to conventional VAE lexicon? The introduction encourages the reader think a VAE mindset, but the MI deviates from VAE convention.
- The author(s) want to improve optimization in the VAE latent space, but spend very little time formulating the optimization procedures in section 4. Lines 159-167 talk about several Bayesian Optimization methods but don't tell the reader which one is used.
- The paragraph spanning 168-183 seemingly discusses how to use a VAE to auxiliary a surrogate design function, yet never really discusses the specifics of how to incorporate the proposed MI estimation other than the vague, "we can further improve the
quality of the candidate set by censoring the moves in latent during gradient ascent that would result
in a value of uncertainty above a predefined threshold."
- The second and third paragraphs of section 5 address my two points above, but are very short on detail. I recommend combing these sections in an effort to provide more mathematical details of the optimization procedure (equations would be nice).
- The proposal to "compute the gradient of the auxiliary property network with respect to latent positions and accept proposal moves along these directions if the decoder uncertainty at the corresponding position in latent is below a predefined threshold (e.g., 99th percentile of decoder uncertainty on the training set" seems problematic if the latent space utilized by the training set is discontinuous, which can and does occur (see, Adversarial Auto-encoders). Are there other ways to incorporate uncertainty that allow for unconnected regions of latent space (other than doing multiple restarts)?

### Significance:
- The applications of this paper are numerous in computational biology and drug design. If it is state-of-the-art, which is unclear (see above), then it is of great significance to researchers in these fields.

**Time Spent Reviewing:**

2

---

> ### Author Response · Authors · 2021-08-10
> **Authors' response to Reviewer VYPR**
>
> Dear reviewer,
>
> We sincerely thank you for your time and valuable feedback. We kindly ask that you read our comment above, addressed to all reviewers, as well as our responses to your specific comments. Below we address your specific comments as best as we can, and we hope you will engage with us actively during the discussion period to clarify any remaining points.
>
> **[I am unfamiliar with the latest and greatest mutual information estimation techniques. However, I do know for certain there is a large body of related research on MI estimation. Could the author(s) please discuss which aspects of their approach are novel? From lines 103-106, it sounds the decomposition (equations 2-4) is not novel. The paper reads as if importance-sampling these quantities is novel. Can the author(s) please confirm this is the case? I think the paper would benefit on a relevant works related to MI estimation alone, since this is a major contribution of the paper.]**
>
> The different concepts that are covered in the Background section (including lines 103-106) are not contributions of our work. Our contributions are summarized in lines 48-55. In particular, the importance sampling estimator (discussed in section 3) is the first important contribution of our work which, as we allude to in the conclusion, can be also leveraged in any other setting involving reliable uncertainty estimation in high-dimensional structured datasets (e.g., active learning, anomaly detection). Regarding your last suggestion, we intended the section 2.2 to play that role: it covers important concepts related to uncertainty quantification / mutual information and the most relevant & up-to-date literature on MI estimation in the settings we focus on in this work (ie. high-dimensional structured data). Was there a particular concept or prior work you thought was missing there?
>
> **[The author(s) mention a variety of related works the first paragraph of section 2.1. Yet, many of these methods do not appear as experimental baselines in section 5. Their absence makes it difficult to assess the proposals of this paper. In particular, how is a researcher to select between existing methods or the ones proposed?]**
>
> Our method can be leveraged in conjunction with any of the VAE-based methods that we discuss in section 2.1 (we will adjust the language in lines 80-82 to be more specific on that particular point). For instance, in section 5, we select two of these methods: the character VAE from Gómez-Bombarelli et al (that operates on molecule SMILES) and the JT-VAE from Jin et al. (that operates on molecular graphs), and demonstrate the value add of our method when used on top of these popular and relatively different architectures for molecular generation.
> Following the reviewer's suggestion, we provide new results comparing the relative performance of models listed in section 2.1, based on the experimental setting discussed in Section 5.3. In particular we achieve SOTA performance on the molecular generation task (please see new results provided in point A in our response to all reviewers above).
>
> **[Conventional VAE lexicon uses x for data space z for latent space. Section 2.2 uses x for inputs and y for discrete outputs. Wouldn't the inputs be y as well since a VAE reconstructs the input from itself? Or is y reconstructed from x? How do x and y in section 2.2 relate to conventional VAE lexicon? The introduction encourages the reader think a VAE mindset, but the MI deviates from VAE convention.]**
>
> Section 2.2 and section 3 are meant to introduce general purpose uncertainty quantification metrics, and purposely exclude any mention of VAEs terminology. We use standard notation accordingly (x for model inputs, y for outputs). This was a conscious decision to facilitate understanding for readers solely interested in the new MI estimator (from section 3), and not by the latent optimization contributions. This comes at the cost of a change of notation between section 3 and section 4 (e.g., for the decoder of a VAE, the inputs are z not x anymore). While notations in section 4 are unambiguous (line 165 for instance refers to M(z) and not M(x) anymore), we will add a sentence to call the reader's attention to this change.
>
> **[- The author(s) want to improve optimization in the VAE latent space, but spend very little time formulating the optimization procedures in section 4. Lines 159-167 talk about several Bayesian Optimization methods but don't tell the reader which one is used. - The paragraph spanning 168-183 seemingly discusses how to use a VAE to auxiliary a surrogate design function, yet never really discusses the specifics of how to incorporate the proposed MI estimation [...] - The second and third paragraphs of section 5 address my two points above, but are very short on detail. I recommend combing these sections in an effort to provide more mathematical details of the optimization procedure (equations would be nice).]**
>
> Section 4 introduces different methods to perform uncertainty-guided optimization in latent space, while section 5 details how these are specifically implemented in our experiments. We believe the separation between section 4 & 5 is important to set apart the conceptual aspects from the implementation details. Please note that all implementation details (e.g., hyperparameters, code) for each experiment are also provided in supplementary material.
> We very much appreciate your feedback on readability though, and have added in section 5 (right below line 212) two algorithms that translate the textual descriptions (lines 197-212) into pseudocode (provided below). Would that properly address your feedback? Could you please clarify the elements you believe would still be missing from section 5 otherwise?
>
> **Bayesian Optimization (Uncertainty censoring) algorithm**
> 1. Uncertainty threshold $T$, number of new points to generate $N$
> 2. Sample $M$ points uniformly at random from the train set, with latent tensor $z$ and property vector $P$
> 3. **For** $i = 1 , N$ **do**
>
>      3.1. Train single task GP on $(z,P)$
>
>      3.2. Generate $B$ candidate points $z_B$ with predicted properties $f_B$ by sequentially maximizing the Expected Improvement
>
>      3.3. Compute decoder uncertainty $MI(z_B)$
>
>      3.4. Set new candidate index $k^* = argmax(f_B)$ s.t. $ MI(z_B) \leq T$
> (or $ k^* =argmin(MI(z_B))$ if constraint not satisfied)
>
>      3.5. Decode new candidate $z_B[k^*]$
>
>      3.6. Obtain true property $p_{k^*}$ of decoded candidate
>
>      3.7. Concatenate $(z_B[k^*], p_{k^*})$ to $(z,P)$
>
>      **End for**
>
> **Gradient ascent algorithm**
> 1. Uncertainty threshold $T$, number of gradient updates $S$, gradient scale $\alpha$
> 2. Sample $M$ points uniformly at random from the train set, with latent tensor $z$ and property vector $P$
> 3. Compute $\nabla_z { h(z)}$, with $h$ the auxiliary network predicting $P$ from $z$
> 4. **For** i = 1 , S **do**
>
>     $z_i = z_i + Alpha * \nabla_z { h(z_i)} $ if $ MI(z_i + \alpha * \nabla_z { h(z_i)}) \leq T $
>
>      **End for**
> 5. Decode final positions z
> 6. Obtain true properties P* of decoded points
>
> **[The proposal to "compute the gradient of the auxiliary property network with respect to latent positions and accept proposal moves along these directions if the decoder uncertainty at the corresponding position in latent is below a predefined threshold (e.g., 99th percentile of decoder uncertainty on the training set" seems problematic if the latent space utilized by the training set is discontinuous, which can and does occur (see, Adversarial Auto-encoders). Are there other ways to incorporate uncertainty that allow for unconnected regions of latent space (other than doing multiple restarts)?]**
>
> The Bayesian optimization-based methods discussed in this work would allow us to swiftly explore the latent space, even if the embedding of the training data in latent space is non-contiguous. We note that, while relatively simple, the uncertainty-constrained gradient ascent method achieves spectacular performance in several experimental settings, in particular achieving SOTA in for molecular generation with JT-VAE.
>
> **[The applications of this paper are numerous in computational biology and drug design. If it is state-of-the-art, which is unclear (see above), then it is of great significance to researchers in these fields.]**
>
> As mentioned above, we obtain SOTA performance on the molecular generation task (point A in our response to all reviewers above). We agree with the reviewer that computational biology (e.g., protein design, antibody affinity optimization) and drug design are two fields that would benefit from the approaches developed in this text. Recently, other teams reached out to us to adapt the methods introduced in this paper to new product development (e.g., cars, software apps). Additionally the MI estimator we introduced can be used in other settings beyond latent optimization (e.g., active learning, anomaly detection) in the high-dimensional structured settings covered in our work or others (e.g., NLP, computer programs).
>
> **[(Limitations and risks) The author(s) do not really address these issues. Can the author(s) please think of times their uncertainty estimation might produce false estimates or when their optimization proposals might avoid less probably but valid regions of latent space that might be associated with a less-represented group?]**
>
> Section F from supplementary material discusses limitations & societal impact. We will move it right before the conclusion in the main text.
>
> We hope that our response and the new experiments have adequately addressed your concerns. We would greatly appreciate it if you could engage with us during the discussion period on any remaining barriers to raising your score.
>
> Thank you,
> The Authors

---

> > ### Comment · Reviewer_VYPR · 2021-08-17
> > **Post-rebuttal reply.**
> >
> > Thank you for your reply. Thank you for providing the requested details. Increasing my score accordingly.

---

> > > ### Author Response · Authors · 2021-08-20
> > > **Post rebuttal response**
> > >
> > > Dear reviewer,
> > >
> > > Thank you for your response and constructive feedback during reviews.
> > >
> > > Kind regards,
> > >
> > > The authors

---

### Official Review · Reviewer_J3j1 · 2021-07-22

**Rating:** 7
**Confidence:** 3

**Summary:**

The authors use importance sampling to estimate the epistemic uncertainty of the decoder, and use this quantity to avoid latent space regions with high uncertainty when generating high-dimensional discrete data.

**Limitations And Societal Impact:**

See my comments above regarding limitations. If published in NeurIPS the authors could include a section about potential ethical/societal implications of their work as a generative modeling method in the camera-ready version of their paper.

**Main Review:**

I believe that if certain concerns are addressed the authors method can be of practical use to important application domains. The paper is in general well-written and well-motivated.

My main concern with the paper is a fundamental one. The authors method corresponds to censoring the regions of the latent space into which not many encodings were made. While this may understandably result in more valid generated samples, is this not against the idea of using variational autoencoders in the first place? After all, when taken to extreme, the authors approach would turn the VAE to a classical autoencoder, where the latent space has no regularity or smoothness outside the already existing encodings. One of the main advantages of using a VAE is (hopefully) obtaining a 'regular' latent space, thereby allowing the creation of samples that are both novel _and_ valid. I understand that this is not always the case in terms of validity (hence the authors' paper) but the route the authors have taken runs the risk of defeating the purpose of using a VAE in the first place. After all, they might be sacrificing novelty for validity, since an autoencoder could also create very valid samples, but with no novelty at all.

A different approach that I believe the authors can compare their approach to is work on robust VAEs - I believe such approaches might be competitive with the authors' in terms of novelty / validity trade-off, remaining truer to the idea of using VAEs. Such approaches strive to create more 'smooth' and 'regular' latent spaces, example work include [1] and [2]. How would the authors compare their work to those who attempt to create a more smooth latent space?

I believe an empirical comparison would also be called for, both for the existing comparisons and a potential comparison with the approaches mentioned above. I think the authors should discuss any potentially useful metric that would characterize the novelty of the generated samples, and compare their work with previous ones in this respect as well. This comparison would especially be useful when made against robust VAE approaches.

I am looking forward to authors' response and am open to revising my score accordingly.

[1] https://arxiv.org/abs/2102.07559
[2] https://arxiv.org/abs/2012.03715

**Time Spent Reviewing:**

4

---

> ### Author Response · Authors · 2021-08-10
> **Authors' response to Reviewer J3j1**
>
> Dear reviewer,
>
> We sincerely thank you for your time and valuable feedback. We kindly ask that you read our comment above, addressed to all reviewers, as well as our responses to your specific comments. Below we address your specific comments as best as we can, and we hope you will engage with us actively during the discussion period to clarify any remaining points.
>
> **[My main concern with the paper is a fundamental one. The authors method corresponds to censoring the regions of the latent space into which not many encodings were made. While this may understandably result in more valid generated samples, is this not against the idea of using variational autoencoders in the first place? After all, when taken to extreme, the authors approach would turn the VAE to a classical autoencoder, where the latent space has no regularity or smoothness outside the already existing encodings. One of the main advantages of using a VAE is (hopefully) obtaining a 'regular' latent space, thereby allowing the creation of samples that are both novel and valid. I understand that this is not always the case in terms of validity (hence the authors' paper) but the route the authors have taken runs the risk of defeating the purpose of using a VAE in the first place. After all, they might be sacrificing novelty for validity, since an autoencoder could also create very valid samples, but with no novelty at all.]**
>
> The constraints that we are imposing during latent optimization are in terms of uncertainty of the decoder and not with respect to distance to the embedding of the training data. Being close to the training data embedding in latent space implies a low decoder uncertainty, but the reverse is not necessarily true. In other words, by imposing constraints on the decoder uncertainty, we are not forcing points to be close to the training data. In fact, in most of our experiments (digit generation and molecular generation) the points maximizing the black box objective were actually very "far" from the training data in latent space (as evidenced by the higher black-box objective values obtained as we increase optimization bounds). A method that would be based on distance to the training data embedding in latent would fare very poorly, unlike our method that is able to properly filter out points leading to invalid decoding, while still identifying valid / high quality objects that maximize the black-box objective.
> We understand the concerns of the reviewer with respect to the fact that overlying additional constraints may potentially result in a lower novelty of the generated objects. To that end, we have expanded our comparisons across baselines including novelty metrics, defined as the % of generated objects that were not seen during training (see point B in our response to all reviewers above). As can be seen, 100% of the generated molecules are novel for our approach.
>
> **[A different approach that I believe the authors can compare their approach to is work on robust VAEs - I believe such approaches might be competitive with the authors' in terms of novelty / validity trade-off, remaining truer to the idea of using VAEs. Such approaches strive to create more 'smooth' and 'regular' latent spaces, example work include [1] and [2]. How would the authors compare their work to those who attempt to create a more smooth latent space?]**
>
> We see our method as complementary to methods that would seek to robustify the latent space. These methods will help lower the % of invalid decodings during latent space optimization, however the constraints imposed during training may artificially constrain the generated objects in non obvious ways. Furthermore, a practitioner may not want to have to re-train an already trained generative model to robustify it, or there might be additional constraints that would not play well with the training procedure suggested in these robust VAE papers -- for example, it would not obvious to extend these methods to Junction Tree VAEs (used in section 5.3.2) since the latent space is used partly for tree decoding, partly for graph decoding. Instead of modifying the VAE training procedure, our method takes as input any already trained VAE model and seeks to improve the latent search optimization directly. As such, it may be used with vanilla VAE architectures or in conjunction with robust VAE approaches. We will make this distinction with existing methods clearer in the background section (section 2.1).
>
> **[I believe an empirical comparison would also be called for, both for the existing comparisons and a potential comparison with the approaches mentioned above. I think the authors should discuss any potentially useful metric that would characterize the novelty of the generated samples, and compare their work with previous ones in this respect as well. This comparison would especially be useful when made against robust VAE approaches.]**
>
> We are extending our comparison with methods discussed in section 2.1 and adding validity metrics comparisons as suggested (see points A and B in our response to all reviewers above).
>
> **[If published in NeurIPS the authors could include a section about potential ethical/societal implications of their work as a generative modeling method in the camera-ready version of their paper.]**
>
> We will move the section F from supplementary material (discussing limitations/societal impact) right before the conclusion in the main text.
>
> We hope that our response and the new experiments have adequately addressed your concerns. We would greatly appreciate it if you could engage with us during the discussion period on any remaining barriers to raising your score.
>
> Thank you,
> The Authors

---

> > ### Author Response · Authors · 2021-08-26
> > **Authors' response to Reviewer J3j1**
> >
> > Dear reviewer,
> >
> > Hope this finds you well. As we are approaching the end of the discussion window (Sep 2nd), we wanted to confirm with you that our response adequately addressed the questions raised in your review.
> >
> > Looking forward to discussing further with you if there are still points to clarify.
> >
> > Kind regards,
> > The authors

---

> > > ### Comment · Reviewer_J3j1 · 2021-08-30
> > > **Thank you for your response**
> > >
> > > Dear authors,
> > >
> > > Thank you for the effort you put in resolving reviewers' comments. I think my concerns were sufficiently addressed and I raise my score accordingly.

---

### Author Response · Authors · 2021-08-10
**Authors' response to all reviewers**

We thank all reviewers for their detailed feedback, and are excited about the new conference guidelines encouraging reviewers to have an active discussion with the authors during the discussion period. We believe this will help resolve remaining misunderstandings in the reviews and our response. Our response is structured as follows:
1. We are providing right below additional results that are directly addressing comments that are common across several reviews.
2. We are also addressing individual reviewers’ comments below each review, and asking clarifying questions where needed.


### A. Detailed performance comparison with prior methods for molecular generation discussed in section 2.1 [Reviewers J3j1 & VYPR]

**Table 1:** We obtain state-of-the-art performance for molecular generation using Junction Tree VAEs [Jin et al., 2019] and gradient ascent if we go sufficiently `far away’ in latent space. The table below reports the penalized logP values for the Top 1 / Top 2 / Top 3 generated molecules with our method and prior approaches discussed in section 2.1.

| Model                                 | Optimization method                             | Top 1 $\uparrow$| Top 2 $\uparrow$ | Top 3 $\uparrow$|
|---------------------------------------|-------------------------------------------------|-------|-------|-------|
| JT-VAE (original) [Jin et al, 2019] | Bayesian Optimization                           | 5.30  | 4.93  | 4.49  |
| MolDQN [Zhou et al., 2019]         | Reinforcement learning                          | 11.84 | 11.84 | 11.82 |
| GraphAF [Shi et al., 2020]         | Reinforcement learning                          | 12.23 | 11.29 | 11.05 |
| CCGF [Liu et al., 2020]        | Chance-constrained optimization                 | 12.32 | 11.79 | 11.61 |
| ChemBO [Korovina et al., 2019]      | Bayesian Optimization                           | 18.39 | -     | -     |
| JT-VAE [Tripp et al., 2020]         | Bayesian Optim. & retraining (median of 5 runs) | 21.20 | 15.34 | 15.34 |
| JT-VAE [Tripp et al., 2020]         | Bayesian Optim. & retraining (best over 5 runs) | 27.84 | 27.59 | 27.21 |
| JT-VAE (ours)                         | Gradient ascent (mean of 10 runs)               | 23.65 | 21.17 | 19.45 |
| JT-VAE (ours)                         | Gradient ascent (best over 10 runs)             | **30.81** | **30.00** | **29.82** |


**Table 2:** We obtain state-of-the-art performance in terms of penalized logP via gradient ascent. However, most generated molecules are of very low quality (only ~1\% pass the quality filters from [Brown et al., 2019]. Leveraging the uncertainty of the decoder (IS-MI) during optimization helps generate molecules with high penalized logP and high quality. NLLP constraints help maintain high quality but lead to suboptimal black-box objective values.

| Decoder uncertainty | Penalized logP - Before filters - Top 1 $\uparrow$ | Penalized logP - Before filters - Avg. top 10 $\uparrow$ | Quality top 10 (\%) $\uparrow$ | Penalized logP - Passing filters - Top 1 $\uparrow$ | Penalized logP - Passing filters - Avg. top 10 $\uparrow$ |
|---------------------|---------------------------------------------------|--------------------------------------------------------|---------------------------------|---------------------------------------------------|---------------------------------------------------------|
| None                | $23.7 \pm 1.3$                                    | $17.0 \pm 0.6$                                         | $ 1 \pm 1 $                 | $1.2 \pm 1.2$                                     | $0.3 \pm 0.3$                                           |
| NLLP                | $3.0 \pm 0.1$                                     | $2.5 \pm 0.1$                                          | $ 82  \pm 6 $                | $3.0 \pm 0.1$                                     | $2.0 \pm 0.2$                                           |
| IS-MI               | $8.4 \pm 10.8$                                    | $6.0 \pm 0.3$                                          | $ 89 \pm 3 $                | $7.7 \pm 0.7$                                     | $5.3 \pm 0.3$                                           |

### B. Measuring performance of our method Vs baselines on additional metrics from the Guacamol benchmark [Reviewer J3j1 & nbh7]

We report additional metrics from the Guacamol benchmark (unicity, novelty, validity) for the molecular generation experiments in the main text. For the JT-VAE gradient ascent experiments, all three metrics were equal to 100% for our importance sampling-based method and the baselines, i.e. all generated molecules were valid, unique (no duplicate among generated molecules) and novel (no generated molecule was present in the training data). Results for the CVAE Bayesian Optimization are reported in the table below.

**Table 3:** CVAE Bayesian Optimization results - Unicity, Novelty & Validity metrics from the Guacamol benchmark

| Bounds | Method | Unicity | Novelty | Validity |
|--------|--------|---------|---------|----------|
| 5      | None   | 100%    | 100%    | 22%      |
| 5       | NLLP   | 100%    | 100%    | 30%      |
| 5       | TI-MI  | 100%    | 100%    | 21%      |
| 5       | IS-MI  | 100%    | 100%    | **33%**      |
| 10     | None   | 100%    | 60%     | 1%       |
| 10      | NLLP   | 100%    | 80%     | 3%       |
| 10     | TI-MI  | 100%    | 80%     | 2%       |
| 10    | IS-MI  | 99%     | **100%**    | **11%**      |
| 15     | None   | 100%    | 64%     | 1%       |
| 15       | NLLP   | 100%    | 64%     | 1%       |
| 15       | TI-MI  | 100%    | 73%     | 1%       |
| 15       | IS-MI  | 96%     | **100%**    | **5%**       |

---

### Decision · Program_Chairs · 2021-09-27

**Decision:**

Accept (Poster)

**Comment:**

Generation of discrete objects from a conditional density (a decoder) with a continuous latent is of interest in many applications.
This work proposes to use  importance sampling to estimate the uncertainty of the decoder, "and use this quantity to avoid latent space regions with high uncertainty when generating high-dimensional discrete data."

The key contribution of the work is an importance-sampling-based estimate to measure decoder uncertainty.
This is used to avoid latent codes that correspond to invalid objects.

The paper is found solid and insightful by the reviewers. During the rebuttal, the authors successfully answered several concerns, and provided a detailed performance comparison with prior methods for molecular generation as well as for measuring performance on additional metric to illustrate the performance of the proposed method.

Overall, there is a consensus for acceptance and I also agree with this decision.